# Structural insights into how DEK nucleosome binding facilitates H3K27 trimethylation in chromatin

Tomoya Kujirai [1,2,8], Kenta Echigoya [1,3,8], Yusuke Kishi [4,5,8], Mai Saeki[4,5], Tomoko Ito[1], Junko Kato[1], Lumi Negishi [1], Hiroshi Kimura [6], Hiroshi Masumoto[7], Yoshimasa Takizawa [1], Yukiko Gotoh [5] ✉ & Hitoshi Kurumizaka [1,2,3] ✉

Structural diversity of the nucleosome affects chromatin conformations and regulates eukaryotic genome functions. Here we identify DEK, whose function is unknown, as a nucleosome-binding protein. In embryonic neural progenitor cells, DEK colocalizes with H3 K27 trimethylation (H3K27me3), the facultative heterochromatin mark. DEK stimulates the methyltransferase activity of Polycomb repressive complex 2 (PRC2), which is responsible for H3K27me3 deposition in vitro. Cryo-electron microscopy structures of the DEK–nucleosome complexes reveal that DEK binds the nucleosome by its tripartite DNA-binding mode on the dyad and linker DNAs and interacts with the nucleosomal acidic patch by its newly identified histone-binding region. The DEK–nucleosome interaction mediates linker DNA reorientation and induces chromatin compaction, which may facilitate PRC2 activation. These findings provide mechanistic insights into chromatin structure-mediated gene regulation by DEK.

Eukaryotic genomic DNA is folded into chromatin, in which the nucleosome is the basic unit. The nucleosome wraps DNA around a histone octamer containing H2A, H2B, H3 and H4 (ref. 1). Chromatin conformations influence cellular processes, such as transcription, replication, repair and recombination[2].

Heterochromatin, characterized by its condensed chromatin state that generally suppresses gene expression, can be categorized into facultative heterochromatin and constitutive heterochromatin[3,4]. Facultative heterochromatin is formed in a context-dependent manner around chromatin regions with developmentally regulated genes[3,4]. In contrast, constitutive heterochromatin is formed in gene-sparse regions, such as pericentromeric and telomeric chromosome regions, across cell types[3,4].

DEK is an abundant nonhistone chromosomal protein with preferential DNA-binding activity to supercoiled and cruciform DNA structures[5–8] and participates in several cellular processes, including RNA splicing, DNA repair and transcription[9,10]. DEK was first identified as a fusion protein with nucleoporin 214 (NUP214), a component of the nuclear pore complex, in a leukemia-associated chromosomal translocation[11]. DEK is widely expressed in proliferating cells, such as progenitor cells, but its production is restricted in differentiated cells that are in a quiescent state, highlighting its potential role in cell proliferation[9].

Supporting this, DEK overexpression is frequently observed in cancer cells with abnormal cell proliferation and micronucleus formation[9,12,13]. Deletion of DEK causes cancer cell death, enhances

[1]Laboratory of Chromatin Structure and Function, Institute for Quantitative Biosciences, The University of Tokyo, Tokyo, Japan. [2]Laboratory for Transcription Structural Biology, RIKEN Center for Biosystems Dynamics Research, Yokohama, Japan. [3]Department of Biological Sciences, Graduate School of Science, The University of Tokyo, Tokyo, Japan. [4]Laboratory of Molecular Neurobiology, Institute for Quantitative Biosciences, The University of Tokyo, Tokyo, Japan. [5]Graduate School of Pharmaceutical Sciences, The University of Tokyo, Tokyo, Japan. [6]Cell Biology Center, Institute of Integrated Research, Institute of Science Tokyo, Yokohama, Japan. [7]Biomedical Research Support Center, Nagasaki University School of Medicine, Nagasaki, Japan. [8]These authors contributed equally: Tomoya Kujirai, Kenta Echigoya, Yusuke Kishi. ✉e-mail: ygotoh@mol.f.u-tokyo.ac.jp; kurumizaka@iqb.u-tokyo.ac.jp

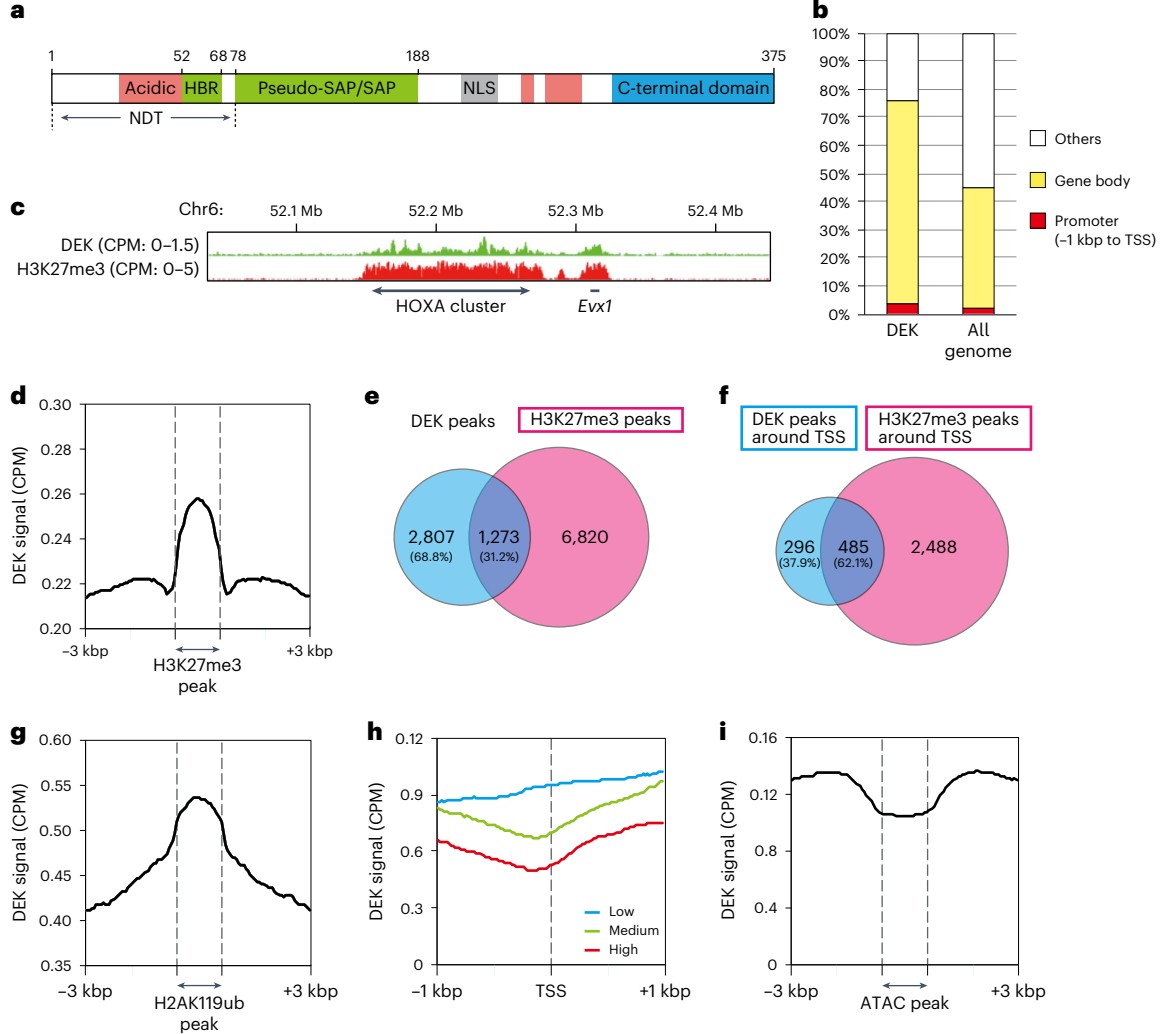

**Fig. 1 | DEK colocalizes with H3K27me3 in mouse NPCs. a**, Schematic representation of DEK with the known regions. NLS, nuclear localization signal. **b**, The proportions of DEK localized in three gene regions (promoter, gene body or others). The gene body includes the exon and intron. All genome indicates the proportion (%) of each category with the distribution on unique sequences of the mouse genome for comparison. **c**, A browser view of the distribution of the DEK and H3K27me3 signals around the HOXA cluster. CPM, counts per million reads.

**d**, DEK signal around H3K27me3 peaks. **e**, Venn diagram of global localizations of the DEK and H3K27me3 peaks. **f**, Venn diagram of localizations of the DEK and H3K27me3 peaks around TSSs. **g**, DEK signal around H2AK119ub peaks. **h**, DEK localization around TSSs in genes with low, medium or high expression. **i**, Global DEK signal around ATAC peaks. These experiments were performed with 13 embryos from 1 pregnant mouse at E11.5 and repeated twice and consistent results were obtained.

the effect of an anticancer drug (tamoxifen) and slows tumor growth[14–16]. Thus, DEK has been suggested as a potential target for cancer treatment[13,16]. However, the mechanism by which DEK affects the chromatin architecture and functions in genome regulation has remained elusive.

In the present study, we identified DEK as a nucleosome-binding protein. Genome-wide localization analyses in mouse embryonic brain demonstrated that DEK is concentrated in facultative heterochromatin, which is marked with H3 K27 trimethylation (H3K27me3). Consistently, H3K27me3 deposition on chromatin is upregulated in the presence of DEK in vitro. The cryo-electron microscopy (cryo-EM) structure of the DEK–nucleosome complex reveals how DEK achieves the dual-tripartite binding to the nucleosomal dyad and linker DNAs and reorients the linker DNA by the sharp kinks at its binding sites on the nucleosomal DNA. Atomic force microscopy (AFM) observation of the DEK–polynucleosome complex establishes that DEK induces chromatin compaction. These findings explain the mechanism by which DEK mediates chromatin compaction and facilitates H3K27me3 deposition through its nucleosome binding.

## Results

### DEK predominantly localizes to facultative heterochromatin

In eukaryotes, nucleosome-binding proteins alter the chromatin conformation and are major regulators of genome function[17–20]. We performed a nucleosome pulldown assay with a HeLa cell nuclear extract and identified the proteins that coprecipitated with the nucleosome beads by mass spectrometry. Among them, we found DEK as a major nucleosome-associating protein. Human DEK is composed of 375 amino acid residues, comprising an N-terminal disordered tail (NDT), the globular pseudo-SAP/SAP domain (hereafter SAP domain), the globular C-terminal domain and the linker region connecting the globular SAP and C-terminal domains[7,21,22] (Fig. 1a).

The ENCODE database showed that DEK is expressed in various mouse tissues, particularly in the central nervous system at embryonic day 11.5 (E11.5) (Extended Data Fig. 1a). We then analyzed the genome-wide localization of DEK in neural progenitor cells (NPCs) of mouse brain at E11 by chromatin immunoprecipitation and sequencing (ChIP-seq) analysis, using an anti-DEK antibody. The genomic features of DEK localization were evaluated by calculating the proportion

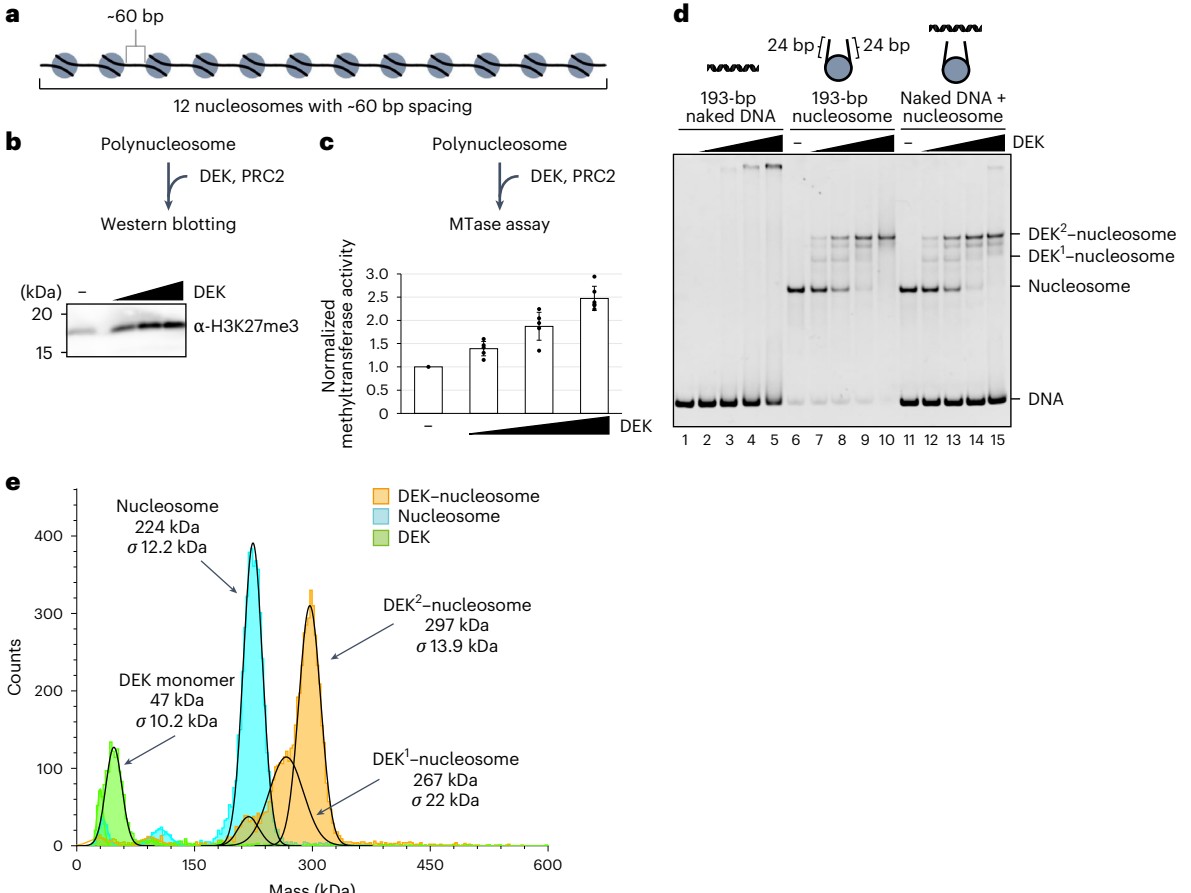

**Fig. 2 | DEK promotes H3K27me3 deposition by PRC2 and specifically binds the nucleosome in vitro. a**, Schematic representation of the polynucleosome with 12 repeats of a 208-bp DNA repeat unit. **b**, Western blot analysis of the H3K27me3 deposition by PRC2 with the 208-bp × 12 polynucleosome, in the presence or absence of DEK, using the anti-H3K27me3 antibody. This experiment was repeated four times and consistent results were obtained. **c**, PRC2 methyltransferase assay with the 208-bp × 12 polynucleosome in the presence or absence of DEK. The metabolite produced by PRC2, *S*-adenosyl ʟ-homocysteine, was measured. The mean (bars) and s.e.m. (error bars) of three experiments are shown. **d**, DEK binding assay to the naked 193-bp DNA or the 193-bp nucleosome

with linker DNAs. Increasing amounts of DEK were mixed with the DNA (lanes 1–5), the nucleosome (lanes 6–10) or both (lanes 11–15). The resulting samples were analyzed by native PAGE with ethidium bromide staining. The bands corresponding to the DEK²–nucleosome and DEK¹–nucleosome complexes are indicated. **e**, Mass photometric analyses of the DEK–nucleosome complex (orange), nucleosome (cyan) and DEK protein (green). The peaks were fit to a sum of Gaussians and the mean of the fitted peaks is shown as the estimated molecular weight for each component. σ represents the s.d. of the fitted Gaussian. This experiment was reproduced independently twice.

of peaks detected in each category (gene and intergenic regions) (Fig. 1b). We found that DEK was specifically localized in the gene regions (~80%). Interestingly, the DEK peaks were especially concentrated in genes related to tissue development, such as pattern specification process, regionalization and nervous system development (Extended Data Fig. 1b).

DEK is reportedly associated with H3K9me3, concomitant with its direct binding to the constitutive heterochromatin protein HP1α (ref. 23). However, the genes required for brain development may be regulated by facultative heterochromatin, in which the H3K27me3 is deposited by the Polycomb repressive complex 2 (PRC2) methyltransferase[4,24–26]. Intriguingly, we found that DEK was localized in the homeobox A (HOXA) cluster, which is suppressed by H3K27me3 (ref. 4; Fig. 1c). HOX genes are reportedly clustered in facultative heterochromatin loci in the genome and activated during developmental processes, including normal embryonic development of the hindbrain and spinal cord[27].

We also found that approximately one third of the DEK peaks (31.2%) were colocalized with the H3K27me3 peaks (Fig. 1d,e). Particularly, around transcription start sites (TSSs), more than half of the DEK peaks (62.1%) overlapped with the H3K27me3 peaks (Fig. 1f).

DEK also colocalized with monoubiquitinated H2AK119 (H2AK119ub), which is deposited by PRC1 ubiquitin ligase (Fig. 1g). H2AK119ub is reportedly essential for transcriptional suppression in facultative heterochromatin[28]. We then performed an RNA sequencing (RNA-seq) analysis and found that DEK was predominantly accumulated around the TSSs of lowly expressed genes (Fig. 1h). Conversely, less DEK was present around the TSSs of highly expressed genes (Fig. 1h). These results suggested that DEK may suppress transcription, probably by altering the chromatin conformation around TSSs. Consistent with this idea, an assay for transposase-accessible chromatin using sequencing (ATAC-seq) analysis revealed the absence of the DEK signal from the open chromatin loci that are frequently found around active TSSs (Fig. 1i). The DEK–H3K27me3 colocalization and the DEK accumulation around the TSSs of lowly expressed genes were also observed in HEK293 human cells (Extended Data Fig. 1c,d). Therefore, DEK may suppress gene expression around TSSs by altering the chromatin conformation, together with H3K27me3 and H2AK119ub deposition.

## DEK stimulates H3K27me3 by PRC2
We next tested whether DEK affects the H3K27me3 deposition by PRC2 in chromatin. To do so, we prepared a recombinant human DEK protein

**Table 1 | Cryo-EM data collection, reconstruction and model refinement parameters**

| | 1. Structure of nucleosome complexed with two DEK molecules (EMD-37115), (PDB 8KCY) | 2. Structure of nucleosome complexed with one DEK molecule (EMD-37121), (PDB 8KD1) | 3. Structure of H1.2 bound to the nucleosome (EMD-37149), (PDB 8KEO) |
|---|---|---|---|
| **Data collection and processing** | | | |
| Magnification | ×81,000 | ×81,000 | ×81,000 |
| Voltage (kV) | 300 | 300 | 300 |
| Electron exposure ($e^-$ per Å$^2$) | 55.2 | 55.2 | 56.0 |
| Defocus range (μm) | −1.0 to −2.5 | −1.0 to −2.5 | −1.0 to −2.5 |
| Pixel size (Å) | 1.05 | 1.05 | 1.05 |
| Symmetry imposed | $C_1$ | $C_1$ | $C_1$ |
| Initial particle images (number) | 4,785,987 | 4,785,987 | 3,603,969 |
| Final particle images (number) | 950,776 | 114,909 | 27,130 |
| Map resolution (Å) | 2.8 | 3.2 | 4.0 |
| FSC threshold | 0.143 | 0.143 | 0.143 |
| Map resolution range (Å) | 2.6–6.9 | 3.0–9.1 | 3.9–11.6 |
| **Refinement** | | | |
| Initial model used (PDB code) | 5LNO | 5LNO | 5LNO, 8HOV |
| Model resolution (Å) | 2.8 | 4.0 | 4.1 |
| FSC threshold | 0.5 | 0.5 | 0.5 |
| Map sharpening $B$ factor (Å$^2$) | 0 | −19.53488 | 0.439364 |
| Model composition | | | |
| Nonhydrogen atoms | 16,120 | 14,213 | 14,130 |
| Protein residues | 1,026 | 893 | 842 |
| Nucleotide | 386 | 346 | 366 |
| $B$ factors (Å$^2$) | | | |
| Protein | 69.27 | 137.15 | 103.31 |
| Nucleotide | 203.34 | 182.14 | 198.91 |
| Root-mean-square deviations | | | |
| Bond lengths (Å) | 0.007 | 0.01 | 0.01 |
| Bond angles (°) | 0.911 | 1.18 | 1.21 |
| **Validation** | | | |
| MolProbity score | 1.75 | 1.84 | 1.5 |
| Clashscore | 8.37 | 8.27 | 9.5 |
| Poor rotamers (%) | 0 | 0 | 0.14 |
| Ramachandran plot | | | |
| Favored (%) | 95.71 | 94.27 | 98.54 |
| Allowed (%) | 4.29 | 5.73 | 1.46 |
| Disallowed (%) | 0 | 0 | 0 |

and a polynucleosome with 12 repeats of a 208-bp Widom 601 DNA repeat unit, which contains about 60 bp of linker DNA between nucleosomes (30 bp on both sides of the nucleosome) (Fig. 2a and Extended Data Fig. 1e). This nucleosome spacing closely resembled that of facultative heterochromatin containing H3K27me3 in cells[29]. We performed the H3K27me3 deposition assay with the PRC2 core enzyme, in the presence or absence of DEK, and found that the PRC2-mediated H3K27me3 deposition was drastically upregulated by increasing amounts of DEK (Fig. 2b). Moreover, the methyltransferase assay revealed that DEK substantially augmented the methyltransferase activity of PRC2 in the presence of the polynucleosome (Fig. 2c). These findings are in good agreement with the genomic colocalization of DEK with H3K27me3. Intriguingly, the PRC2-mediated H3K27me3 deposition was not efficient with the mononucleosome as compared to the polynucleosome

(Extended Data Fig. 1f). In addition, our pulldown assay revealed that DEK does not directly bind to PRC2 (Extended Data Fig. 1g,h). Therefore, the higher-order chromatin conformation modulated by DEK may stimulate the H3K27me3 deposition.

## DEK directly binds to the nucleosome

To study the mechanism by which DEK binding influences the chromatin conformation, we conducted a gel mobility shift assay to examine the DEK–nucleosome interaction. We prepared the nucleosome containing the 193-bp Widom 601 DNA, in which the nucleosome is positioned at the center of the DNA, resulting in 24-bp linker DNAs at both ends[30,31] (Extended Data Fig. 1i,j). Consistent with the previous studies, DEK bound to naked DNA but its affinity was low[5,7,8] (Fig. 2d, lanes 1–5). In contrast, DEK robustly bound to the nucleosome with linker DNAs

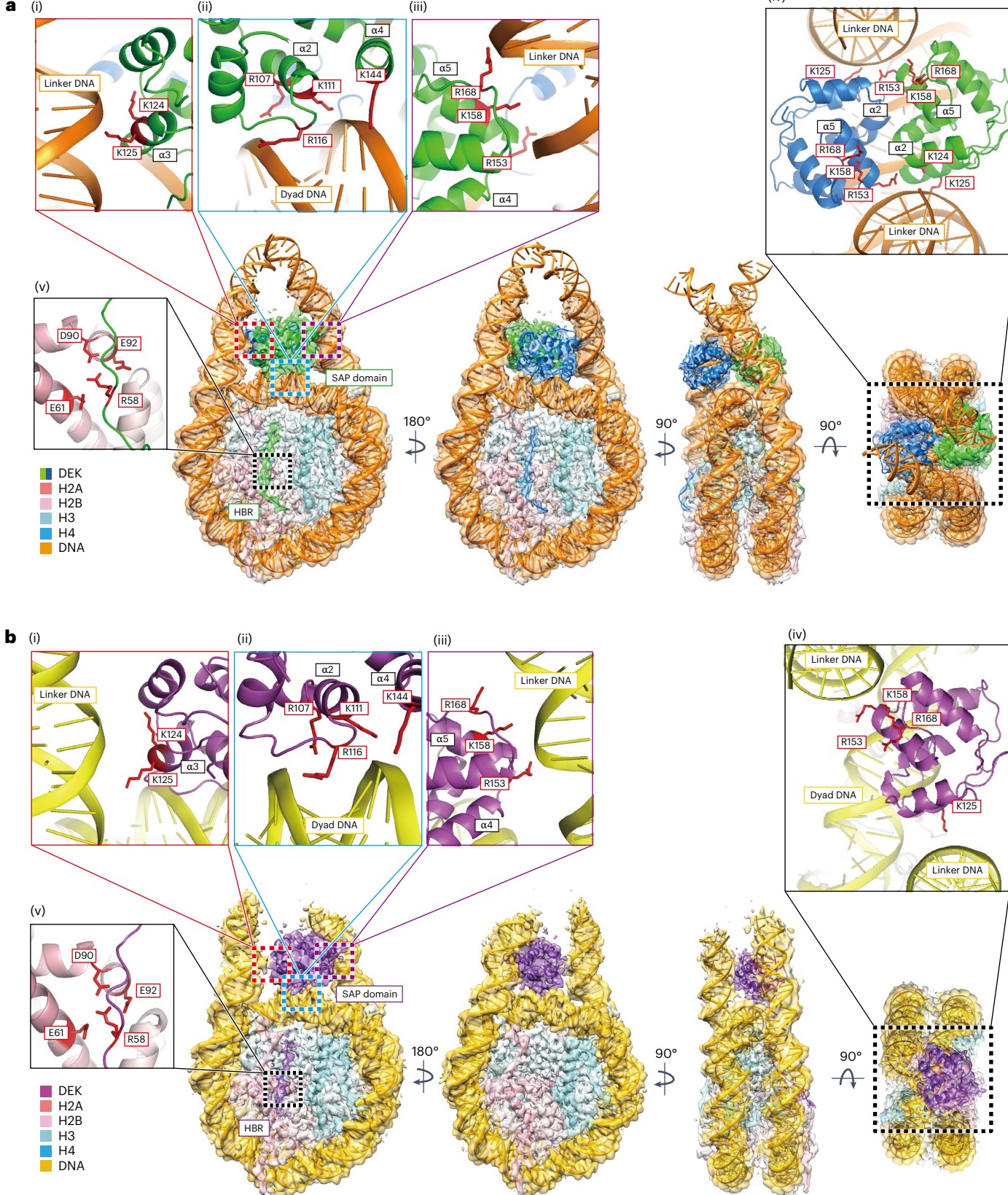

**Fig. 3 | Cryo-EM structures of DEK–nucleosome complexes. a**, Structure of the DEK[2]–nucleosome complex with the EM density map. Close-up views (encircled by dashed boxes) of the DEK regions interacting with the linker DNA on the left side (i), the dyad DNA (ii), the linker DNA on the right side (iii), another DEK molecule on the dyad DNA (iv) and the histone acidic patch surface (v). **b**, Structure of the DEK[1]–nucleosome complex with the EM density map.

Close-up views (encircled by dashed boxes) of the DEK regions interacting with the linker DNA on the left side (i), the dyad DNA (side view) (ii), the linker DNA on the right side (iii), the dyad DNA (top view) (iv) and the histone acidic patch surface (v). The DEK residues contacting the DNA backbone and the nucleosomal acidic patch are highlighted in red.

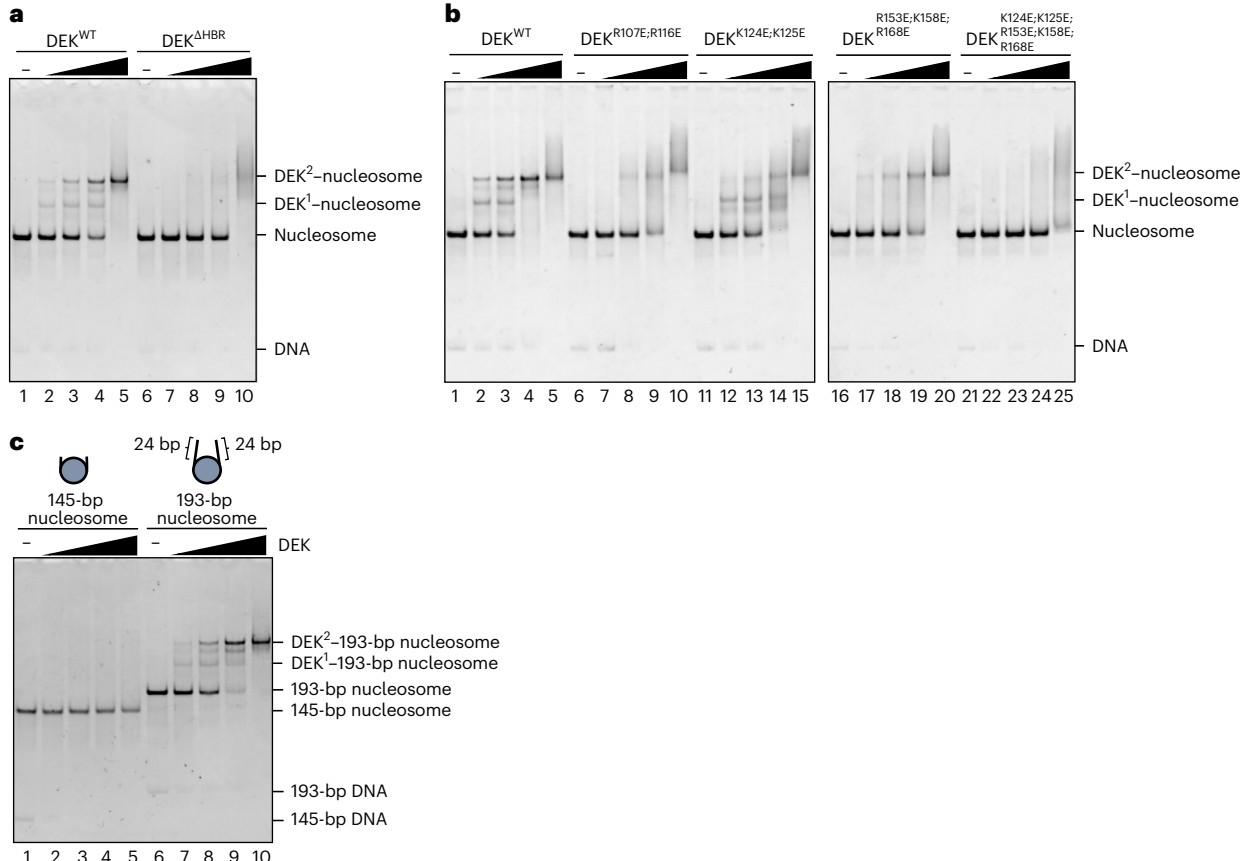

**Fig. 4 | Mutational analysis of DEK nucleosome binding. a**, Gel mobility shift assay of DEK^WT (lanes 1–5) and DEK^ΔHBR (lanes 6–10) with the 193-bp nucleosome. **b**, Gel mobility shift assay of DEK^WT (lanes 1–5), DEK^R107E;R116E (lanes 6–10), DEK^K124E;K125E (lanes 11–15), DEK^R153E;K158E;R168E (lanes 16–20) and DEK^K124E;K125E;R153E;K158E;R168E (lanes 21–25) with the 193-bp nucleosome. **c**, Gel mobility shift assay of DEK to the nucleosome with and without linker DNAs. Increasing amounts of DEK were mixed with the nucleosome without linker DNAs (145-bp nucleosome, lanes 1–5) or with 24-bp linker DNAs (193-bp nucleosome, lanes 6–10). The resulting samples were analyzed by native PAGE with ethidium bromide staining. These results were reproduced in another independent experiment.

(Fig. 2d, lanes 6–10). Interestingly, we detected major slow-migrating and minor fast-migrating bands, which may correspond to two DEK molecules and one DEK molecule bound to a nucleosome, respectively. The mass photometric analysis also identified nucleosomes complexed with one or two molecules of DEK and revealed that the binding of two DEK molecules is a major nucleosome-binding form (Fig. 2e). In addition, the competition assay revealed that DEK specifically binds to the nucleosome over the naked DNA (Fig. 1d, lanes 11–15).

## Cryo-EM structures of the DEK–nucleosome complexes

We determined the cryo-EM structures of the DEK–nucleosome complexes with 24-bp linker DNAs (Table 1, Extended Data Figs. 2–4, and Supplementary Video 1). Consistent with the gel mobility shift assay and the mass photometric analysis, we obtained two structures, the complex with two DEK molecules bound to the nucleosome (DEK^2–nucleosome) and that with one DEK molecule bound to the nucleosome (DEK^1–nucleosome), at 2.8-Å and 3.2-Å resolutions, respectively (Fig. 3a,b). The cryo-EM three-dimensional (3D) classification analysis revealed that the DEK^2–nucleosome is the major nucleosome-binding form, consistent with the gel mobility shift assay and the mass photometric analysis (Fig. 2d,e and Extended Data Fig. 2).

In the DEK^2–nucleosome structure, two SAP domains (residues 78–188) sit nearly symmetrically on the nucleosomal dyad DNA and bind to the dyad DNA and both sides of the linker DNAs (Fig. 3a, (i)–(iv)). Consequently, the two DEK molecules are tightly bound to the nucleosome in the dual-tripartite DNA-binding form (two symmetric DEK SAP domains in a tripartite DNA-binding form). These tripartite interactions between DEK and the dyad DNA and two linker DNAs were also maintained in the DEK^1–nucleosome complex (Fig. 3b, (i)–(iv)).

In the DEK^2–nucleosome structure, one DEK SAP domain contacts the other DEK SAP domain, with the interface formed by the α2 (residues 102–108) and α5 (residues 156–164) helices (Fig. 3a, (iv)). We prepared the DEK^102–105A mutant, in which the residues in the major DEK–DEK contact site between the SAP domains (amino acid residues 102–105 in the α2 helix) were replaced by alanine (Extended Data Fig. 5a,b). Gel mobility shift assays revealed that the DEK^102–105A mutant formed the bands corresponding to the DEK^2–nucleosome and DEK^1–nucleosome complexes as efficiently as wild-type (WT) DEK (Extended Data Fig. 5c). These results suggested that the interactions between the SAP domains in the DEK^2–nucleosome complex may not be critical for the nucleosome binding by DEK.

In both the DEK^2 and DEK^1 complexes, the DEK amino acid residues 52–68 interact with the histone octamer surface, including the acidic patch (Fig. 3a,b, (v)). This DEK region (residues 52–68) was named the histone-binding region (HBR) (Fig. 1a). The R58 residue of the DEK HBR interacts with the histone acidic patch residues, H2A E61, D90 and E92, and may function as an 'arginine anchor', which is often found in nucleosome-binding proteins[19] (Fig. 3a,b, (v)). To test whether the DEK HBR functions in nucleosome binding, we prepared the DEK mutant, DEK^ΔHBR, which lacked residues 52–68 (Extended Data Fig. 5d). Interestingly, DEK^ΔHBR was substantially defective in nucleosome binding (Fig. 4a). This indicated that the HBR has an important role in the nucleosome binding by DEK and may contribute to enhance its specificity to the nucleosome.

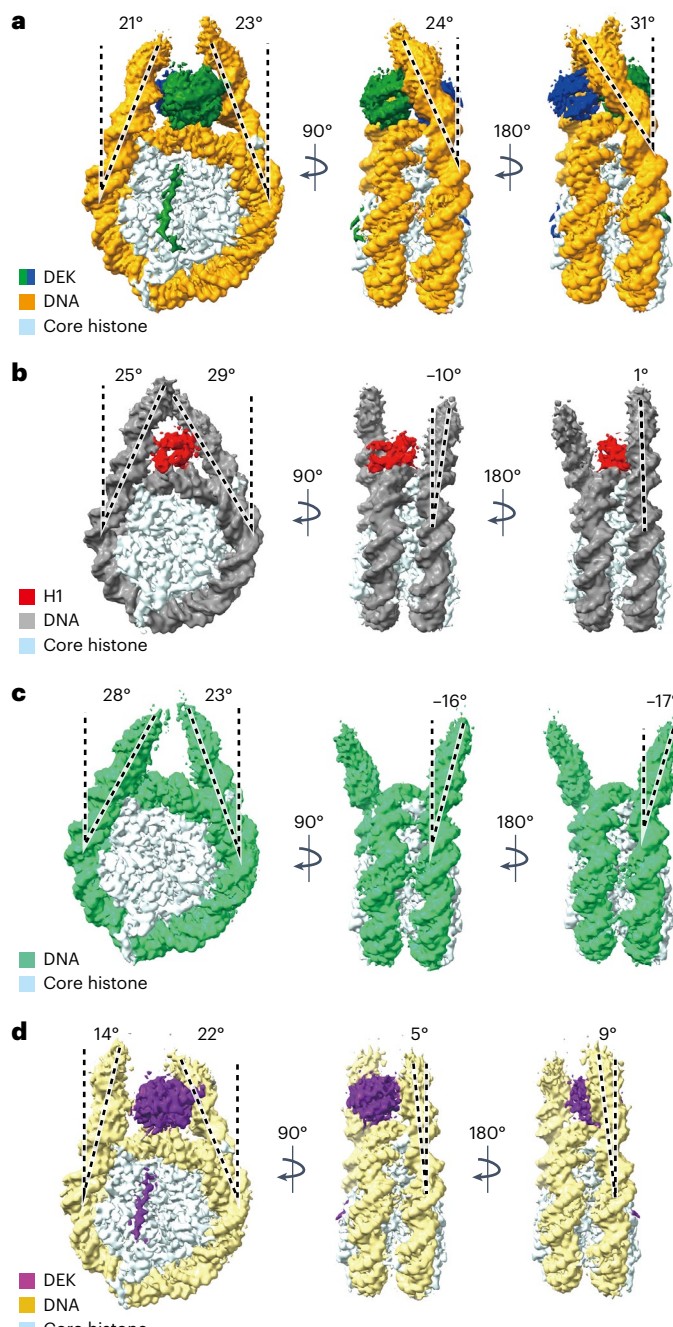

**Fig. 5 | Linker DNA orientations of the nucleosome complexes.**
a–d, Comparison of the linker DNA orientations of the DEK²–nucleosome (a), H1–nucleosome (b), free nucleosome^closed (c) and DEK¹–nucleosome (d) complexes. The angles of the linker DNA in parallel and perpendicular directions are shown. The cryo-EM maps of each complex are presented.

### DEK–DNA interactions are required for nucleosome binding

In the DEK²–nucleosome structure, the SAP α3 helices of DEK interact with a linker DNA through the K124 and K125 residues (Fig. 3a, (i)). The other linker DNA is captured by the DEK R153, K158 and R168 residues of the SAP α5 helix and L5 loop in the same DEK molecule (Fig. 3a, (iii)). These linker DNA interactions of one DEK are symmetrically maintained in the other DEK (Fig. 3a, (iv)). On the dyad DNA, the R107, K111, R116 and K144 residues of one DEK interact with the DNA backbone strand and the other DEK binds to the opposite DNA strand in a symmetrical manner (Fig. 3a, (ii) and (iv), and Extended Data Fig. 5e). These DEK–DNA interactions are mostly conserved in the DEK¹–nucleosome complex,

except for the 22° tilting of the DEK molecule bound to the dyad DNA in the DEK²–nucleosome complex (Extended Data Fig. 5f).

To test whether these DEK–DNA interactions are actually involved in the DEK–nucleosome binding, we prepared three DEK mutants, DEK^K124E;K125E, DEK^R153E;K158E;R168E and DEK^R107E;R116E, in which K124 and K125, R153, K158 and R168 and R107 and R116 were replaced by glutamic acid, respectively (Extended Data Fig. 5g). Analyses of the nucleosome-binding activities of DEK^R153E;K158E;R168E and DEK^R107E;R116E revealed that the band corresponding to the DEK²–nucleosome complex was drastically reduced and the band corresponding to the DEK¹–nucleosome complex disappeared (Fig. 4b, lanes 6–10 and 16–20). These results indicate that the dyad DNA binding by the R107 and R116 residues and the linker DNA binding by the R153, K158 and R168 residues are important for the nucleosome binding by DEK. Because the DEK²–nucleosome complex was still observed with both mutants, while the DEK¹–nucleosome complex was not, the nucleosome binding by two DEK molecules may stabilize the DEK–nucleosome interaction. In contrast, the DEK^K124E;K125E mutant, in which other linker DNA-binding residues, K124 and K125, are substituted, showed somewhat reduced nucleosome-binding activity but its binding profiles as the DEK¹–nucleosome and DEK²–nucleosome forms were unchanged (Fig. 4b, lanes 11–15). However, the nucleosome binding of DEK was drastically decreased when both sets of linker DNA-binding residues, the DEK K124, K125, R153, K158 and R168 residues, were simultaneously replaced by glutamic acid in the DEK^K124E;K125E;R153E;K158E;R168E mutant (Fig. 4b, lanes 21–25). Consistently, DEK did not bind to the nucleosome core particle lacking linker DNAs (Fig. 4c, lanes 6–10, and Extended Data Fig. 5h,i).

DEK has an additional C-terminal DNA-binding domain, which is not visualized in the DEK–nucleosome structures (Figs. 1a and 3). To test whether the C-terminal domain is involved in nucleosome binding, we prepared the DEK^ΔC mutant, which lacks the C-terminal domain (residues 309–375), and tested its nucleosome binding (Extended Data Fig. 5j,k). DEK^ΔC generated the major and minor bands, corresponding to the DEK²–nucleosome and DEK¹–nucleosome, respectively (Extended Data Fig. 5l). The nucleosome-binding activity of DEK^ΔC may be somewhat reduced, as judged from the amount of free nucleosome. Therefore, the DEK C-terminal domain has an auxiliary role in nucleosome binding. Overall, these structural and biochemical findings showed that the tripartite DNA binding by DEK in the dyad and linker DNAs has important roles in its specific nucleosome binding.

### DEK induces kinks in the linker DNAs

Linker histones are major nucleosome-binding proteins, which mediate chromatin compaction, and approximately 50–130% amounts of linker histones relative to nucleosomes are present in cells[32,33]. We determined the cryo-EM structures of the nucleosome and the H1–nucleosome complex with 24-bp linker DNAs and compared them to the structures of the DEK–nucleosome complexes (Fig. 5, Tables 1 and 2, and Extended Data Figs. 6–9). We found that the linker DNAs are sharply bent in the DEK²–nucleosome complex and their orientations shift toward opposite sides compared to those in the free and H1–nucleosome complexes (Fig. 5a–c, side views). Therefore, the binding of two DEK molecules induces substantial linker DNA bending in the nucleosome, which may lead to topological changes in the nucleosomal DNA. Consistent with this idea, DEK binding reportedly induced positive supercoils to chromatin[34]. In contrast, the linker DNA orientation of the DEK¹–nucleosome complex is intermediate between those of the H1–nucleosome and DEK²–nucleosome complexes (Fig. 5a,b,d). In addition, the free nucleosome structures revealed the multiple orientations of the linker DNA, reflecting its inherent flexibility, as reported previously[35–37] (Fig. 5c, Table 2 and Extended Data Fig. 7j,k). In contrast, multiple orientations of the linker DNAs were not observed in the DEK–nucleosome complexes, suggesting that DEK reduces the linker DNA flexibility (Extended Data Fig. 2).

**Table 2 | Cryo-EM data collection and reconstruction parameters**

| | 4. 193-bp Widom 601L nucleosome (closed) (EMD-38443) | 5. 193-bp Widom 601L nucleosome (middle) (EMD-38444) | 6. 193-bp Widom 601L nucleosome (open) (EMD-38445) |
|---|---|---|---|
| **Data collection and processing** | | | |
| Magnification | ×81,000 | ×81,000 | ×81,000 |
| Voltage (kV) | 300 | 300 | 300 |
| Electron exposure (e⁻ per Å²) | 57.2 | 57.2 | 57.2 |
| Defocus range (µm) | −1.0 to −2.5 | −1.0 to −2.5 | −1.0 to −2.5 |
| Pixel size (Å) | 1.06 | 1.06 | 1.06 |
| Symmetry imposed | $C_1$ | $C_1$ | $C_1$ |
| Initial particle images (number) | 5,097,351 | 5,097,351 | 5,097,351 |
| Final particle images (number) | 424,494 | 259,849 | 242,120 |
| Map resolution (Å) | 3.4 | 3.7 | 3.9 |
| FSC threshold | 0.143 | 0.143 | 0.143 |
| Map resolution range (Å) | 3.3–13.4 | 3.5–13.4 | 3.6–15.5 |
| Map sharpening $B$ factor (Å²) | 0 | 0 | 0 |

## DEK induces chromatin compaction

We then analyzed the chromatin conformation of a reconstituted poly-nucleosome containing DEK by AFM. The polynucleosomes were recon-stituted using 12 repeats of the 208-bp Widom 601 DNA sequence[38] (Fig. 6a and Extended Data Fig. 10a,b). The conformations of the poly-nucleosome in the presence or absence of DEK were visualized by AFM (Fig. 6b). The area occupied by each polynucleosome was quantified in the AFM images and displayed as a violin plot (Fig. 6c). The average size of the DEK-containing polynucleosomes was 371 pixels, while that of the free polynucleosomes was 436. These results indicated that DEK significantly compacts the polynucleosome structure. For comparison, we visualized the polynucleosome in the presence of linker histone H1 (Fig. 6b,c). The average size of the H1-containing polynucleosomes was 312 pixels. Therefore, the compaction of the polynucleosome by DEK may not be as drastic as that by linker histone H1.

## Discussion

DEK, identified as an abundant chromatin-associated oncoprotein, is proposed to have roles in various cellular processes, especially transcriptional regulation[9]. However, the mechanism by which DEK regulates transcription in cells has not been clarified so far. Our genome-wide analysis revealed that DEK accumulates in lowly expressed genes and colocalizes with facultative heterochromatin marks, H3K27me3 and H2AK119ub (Fig. 1). These findings suggest that DEK probably suppresses transcription by fostering a sup-pressive chromatin conformation as facultative heterochromatin. Supporting this hypothesis, DEK reportedly restricts H3.3 incorpo-ration, a process associated with transcriptionally active chromatin, in heterochromatin[39,40]. The mechanism of DEK-mediated transcrip-tion suppression has also been suggested to involve the restriction of chromatin accessibility by transcription machinery[41], the inhibition of the p300-dependent and PCAF-dependent histone acetylation[42] and the promotion of histone deacetylation by recruiting a histone deacetylase complex[43]. Along with our results, these facts suggest that DEK probably suppresses gene transcription by forming a suppressive chromatin conformation.

How does DEK form suppressive chromatin? In the present study, we found that one or two molecules of DEK specifically bind to the nucleosome through tripartite DNA binding concomitant with acidic patch binding. This presents the unique instance of a nucleosome-binding mode in which three sections of the DNA, the dyad and linker DNAs, and the acidic patch of the nucleosome

simultaneously bind to a nucleosome-binding protein. Linker histone H1 is a major nucleosome-binding protein that also adopts tripartite DNA binding in the nucleosome but does not interact with the acidic patch[44,45]. Surprisingly, the binding of two DEK molecules to the nucleo-some substantially bends the linker DNAs and compacts the chromatin conformation (Figs. 3, 5 and 6). This remarkable linker DNA bending found in the DEK²–nucleosome complex has not been reported for other proteins complexed with the nucleosome. The DEK-mediated linker DNA reorientation may induce chromatin compaction, which could be responsible for the suppression of genes in facultative het-erochromatin. The PRC2 activity may be regulated by the chromatin structure, in which two nucleosomes are bridged by PRC2 to deposit H3K27me3 on the neighboring nucleosome in chromatin[46–48]. DEK may form a spatial arrangement of two nucleosomes suitable for PRC2 acti-vation by compacting the chromatin. It should be noted that a previous study reported that the loss of DEK did not influence the H3K27me3 levels in human cultured cells[43] and we obtained a consistent result in *Dek*-knockout (KO) mice (Extended Data Fig. 10c–g). This discrepancy may be explained by the possibly redundant functions of DEK and linker histone H1 in chromatin compaction, which may be required for the PRC2 activity. Supporting this idea, H1 also reportedly com-pacts chromatin and facilitates H3K27me3 deposition by PRC2 (refs. 33,46,49,50). Unveiling the functional distinctions and similarities between DEK and H1 in heterochromatin formation will be an interest-ing future issue to study.

DEK overexpression may be related to oncogenesis promotion[14]. In the present study, we found that DEK strongly binds to the nucleosome over the naked DNA and induces substantial DNA bending at the linker DNAs. In DEK-overproducing cells, excess DEK may inappropriately bind nucleosomes, compete with linker histones and compact chro-matin by bending linker DNA. This unusual chromatin reorganization may lead to aberrant H3K27me3 deposition by PRC2 and induce serious gene dysregulation.

The DEK–NUP214 fusion protein is reportedly produced in leu-kemia cells[11,51]. In DEK–NUP214, the NUP214 portion is connected to the DEK C terminus, with the deletion of 26 amino acid residues[11]. Therefore, the DEK regions required for nucleosome binding are maintained in the fusion protein (Extended Data Fig. 10h). In hemat-opoietic progenitor cells, DEK–NUP214 production inhibits differentia-tion, probably by upregulating HOX genes, which may be suppressed by the DEK-mediated heterochromatin formation[52]. Because our genome-wide analysis revealed that DEK colocalizes with HOX genes,

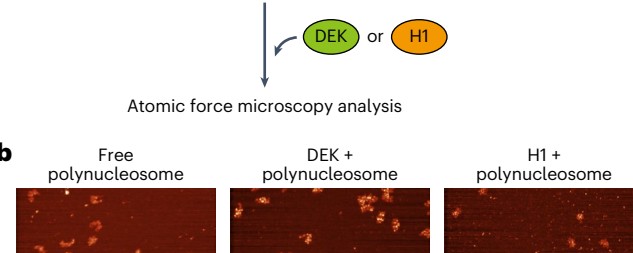

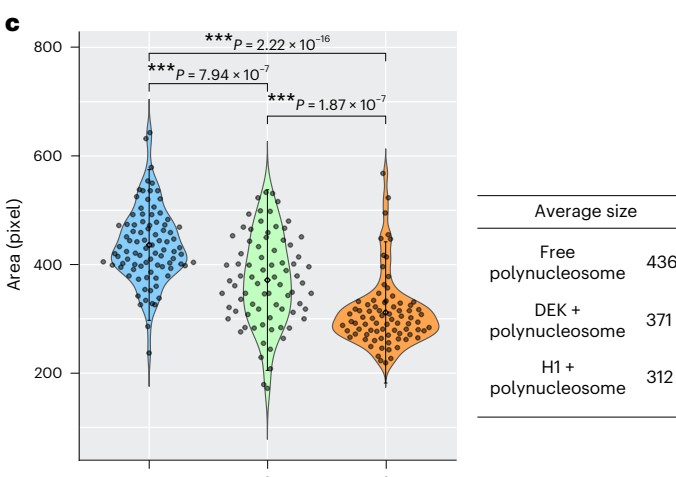

**Fig. 6 | Chromatin compaction by DEK. a**, Schematic representation of the AFM analysis of the polynucleosomes. The polynucleosome with 12 repeats of a 208-bp DNA repeat unit was incubated with DEK and subjected to AFM imaging. **b**, Representative AFM images of polynucleosomes in the absence and presence of DEK or H1. **c**, Violin plots of quantified areas of individual polynucleosome particles shown in **b**. The average sizes (black points) ± 2 s.d. (error bars) of the indicated particles are shown. The $P$ values were determined using the Mann–Whitney $U$-test with a two-sided alternative. ***$P < 0.001$. The average sizes are indicated in the table.

the DEK dysfunction by NUP214 conjugation may be responsible for the HOX gene dysregulation in the cells. Further studies are awaited to tackle this issue.

## Online content

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

## Methods

### Ethics statement

All animals were maintained and studied according to protocols approved by the Animal Care and Use Committee of The University of Tokyo (approval numbers P25-8 and P30-4 in the Graduate School of Pharmaceutical Sciences and 0421 and A2022IQB001 in the Institute for Quantitative Biosciences). All procedures were followed in accordance with The University of Tokyo guidelines for the care and use of laboratory animals and the ARRIVE guidelines.

### Preparation of histone and DEK proteins

The human core histones, H2A, H2B, H3.1 and H4, were fused with a His$_6$ tag at the N terminus. The core histones were expressed in *Escherichia coli* cells and purified using Ni-NTA resin (Qiagen). After tag cleavage, the resulting histone proteins were further purified by chromatography on a MonoS column (Cytiva). The eluted samples were dialyzed against water, lyophilized[53] and stored at 4 °C. The human linker histone H1.2 was fused with a His$_6$–SUMO tag at the C terminus. The linker histone H1.2 was expressed in *E. coli* cells and purified using Ni-NTA resin (Qiagen). After the tag cleavage, the resulting protein was further purified by chromatography on a MonoS column (Cytiva). The eluted sample was dialyzed against H1 buffer (20 mM Tris-HCl pH 7.5, 100 mM NaCl, 10% glycerol and 2 mM 2-mercaptoethanol), frozen in liquid nitrogen and stored at −80 °C (ref. 54). The DNA fragments encoding WT human DEK and its mutants were inserted into the pET21b vector and expressed as an N-terminally His$_6$–SUMO-tagged protein in *E. coli* Rosetta-gami B (DE3) cells. The cells were cultured at 37 °C until the optical density at 600 nm reached 0.6 and then induced to produce the recombinant protein by adding 0.5 mM IPTG and incubating them further at 18 °C overnight. The cells were harvested by centrifugation and resuspended in lysis buffer (50 mM Tris-HCl pH 8.0, 500 mM NaCl, 5% glycerol, 5 mM imidazole, 2 mM 2-mercaptoethanol and 1 mM PMSF). The resuspended cells were disrupted by sonication and the resulting lysate was separated by centrifugation. The tagged DEK protein in the supernatant was subjected to Ni-NTA affinity chromatography (Qiagen) and eluted in lysis buffer containing 500 mM imidazole. The His$_6$–SUMO tag was removed by human rhinovirus 3C protease treatment during dialysis in dialysis buffer (20 mM Tris-HCl pH 7.5, 200 mM NaCl, 5% glycerol, 1 mM EDTA, 2 mM 2-mercaptoethanol and 1 mM PMSF) at 4 °C. The resulting sample was purified by MonoS 5/50 GL (Cytiva) cation-exchange column chromatography with a linear gradient of NaCl in MonoS buffer (50 mM Tris-HCl pH 7.5, 200 mM NaCl, 5% glycerol, 1 mM EDTA, 1 mM PMSF and 2 mM 2-mercaptoethanol) and MonoS buffer containing 600 mM NaCl. The purified DEK protein was dialyzed against dialysis buffer (20 mM Tris-HCl pH 7.5, 150 mM NaCl, 10% glycerol, 0.5 mM EDTA, 0.1 mM PMSF and 1 mM DTT), frozen in liquid nitrogen and stored at −80 °C.

### Preparation of DNA fragments

The 193-bp and 145-bp Widom 601 DNA fragments were cloned into plasmids. The plasmid encoding each DNA fragment was amplified in *E. coli* cells and the DNA was prepared by EcoRV digestion[30,31,55].

For the 193-bp palindromic Widom 601L sequence, a derivative of the left half of the Widom 601 sequence[30], the DNA fragment was purified as follows[56]. Briefly, the DNA fragments containing the 601L(A) or 601L(T) sequence were inserted into the pGEM-T Easy vector (Promega). These DNA fragments contain a HinfI site to produce a 3-nt overhang at one 5′ end (601L(A): 5′-ATT-3′, 601L(A): 5′-AAT-3′). The plasmids were amplified in *E. coli* cells separately. The 601L(A) and 601L(T) fragments were excised by EcoRV and then dephosphorylated by calf intestine alkaline phosphatase. After dephosphorylation, the 3-nt overhang was produced by HinfI digestion. The resulting DNA fragments were ligated to generate the palindromic 193-bp DNA fragment. The sequences are as follows:

The 193-bp 601L DNA, 5′-ATCACGTAATATTGGCCAGCTAGGAT CACAATCCCGGTGCCGAGGCCGCTCAATTGGTCGTAGACAGCTCTAG CACCGCTTAAACGCACGTACGGAATCCGTACGTGCGTTTAAGCGGT GCTAGAGCTGTCTACGACCAATTGAGCGGCCTCGGCACCGGGATTGT GATCCTAGCTGGCCAATATTACGTGAT-3′

The DNA fragment containing 601L DNA (A), 5′-GATATCACG TAATATTGGCCAGCTAGGATCACAATCCCGGTGCCGAGGCCGCT CAATTGGTCGTAGACAGCTCTAGCACCGCTTAAACGCACGTACG GAATCCGGATCCGATATC-3′

The DNA fragment containing 601L DNA (T), 5′-GATATCACG TAATATTGGCCAGCTAGGATCACAATCCCGGTGCCGAGGCCGCT CAATTGGTCGTAGACAGCTCTAGCACCGCTTAAACGCACGTACG GATTCCGGATCCGATATC-3′.

The DNA fragment containing 12 208-bp repeats of the Widom 601 sequence was prepared as described previously[38,57]. Briefly, the plasmid encoding each DNA fragment was amplified in *E. coli* cells and the fragment was isolated by EcoRV digestion, precipitated with polyethylene glycol and further purified on a TSKgel DEAE-5PW column (Tosoh). The 208-bp DNA fragment, representing the mononucleosome unit DNA, was prepared by ScaI digestion of the 12-repeat DNA fragment.

### Nucleosome reconstitution

The histone octamer was reconstituted as described previously[53]. Equimolar amounts of lyophilized histones were dissolved in denaturing buffer and dialyzed in refolding buffer to form the histone octamer. The resulting histone octamer was purified by gel filtration chromatography on a Superdex 200 column (Cytiva).

The nucleosomes were reconstituted with the histone octamer and the indicated DNA fragment by the salt dialysis method[53]. The reconstituted nucleosomes were purified by nondenaturing 6% PAGE, using a Prep Cell apparatus (Bio-Rad). The purified nucleosomes were dialyzed against nucleosome storage buffer (20 mM Tris-HCl pH 7.5, 1 mM DTT and 5% glycerol), flash-frozen in liquid nitrogen and stored at −80 °C.

The polynucleosomes were reconstituted with the histone octamer and the 208-bp DNA fragments containing the 12 repeats of the Widom 601 sequence DNA by the salt dialysis method. After the salt dialysis, aggregates were removed by centrifugation. The resulting samples were further dialyzed against polynucleosome storage buffer (10 mM Tris-HCl pH 7.5, 1 mM DTT and 5% glycerol), flash-frozen in liquid nitrogen and stored at −80 °C (refs. 38,57).

### DEK binding assay

The indicated amount of the DEK protein was mixed with the nucleosome containing the 193-bp DNA (0.1 μM) in 10 μl of reaction buffer (14 mM Tris-HCl pH 7.5, 0.7 mM DTT, 6% glycerol, 75 mM NaCl, 0.25 mM EDTA, 0.05 mM PMSF, 0.001% polyglutamic acid and 0.1 mg ml$^{-1}$ BSA) and incubated at 37 °C for 30 min. The reaction mixtures were subjected to nondenaturing PAGE. The gel was visualized by ethidium bromide staining.

### Mass photometry analyses

To prepare the DEK–nucleosome sample, the 193-bp 601 nucleosome (1 μM) and DEK (2 μM) were mixed in 20 μl of mass photometry buffer (18 mM Tris-HCl pH 7.5, 0.4 mM DTT, 3% glycerol, 60 mM NaCl, 0.3 mM EDTA and 0.02 mM PMSF) and incubated overnight at room temperature. To prepare the samples with free nucleosome or DEK only, the 193-bp 601 nucleosome (1 μM) or DEK (4 μM) was incubated overnight in 20 μl of mass photometry buffer at room temperature. Mass photometry analyses were performed on coverslips coated with 0.01% polylysine (Sigma, P4832, readymade; Sigma, P2658, powder) and recorded on a TWO$^{MP}$ (Refeyn) mass photometer[58]. For data acquisition, each sample was diluted 1:40 in 20 μl of mass photometry buffer on a sample cassette and recorded for 60 s using AcquireMP (Refeyn, version 2023 R1.1). The resulting histograms were fitted to Gaussian

curves and the estimated peak mass was extracted using DiscoverMP (Refeyn, version 2023 R1.2).

## Cryo-EM sample preparations

The preparation of the DEK–nucleosome complex was performed as follows. The 193-bp 601L nucleosome (1.5 µM) and DEK (3.0 µM) were mixed in 410 µl of reaction buffer (20 mM Tris-HCl pH 7.5, 1 mM DTT, 9% glycerol, 122 mM NaCl, 0.4 mM EDTA and 0.08 mM PMSF) and incubated at 37 °C for 30 min to form the complex. The DEK–nucleosome was stabilized using the gradient fixation method[59]. The sucrose gradient was prepared using sucrose buffer (10 mM HEPES–NaOH pH 7.5, 20 mM NaCl, 1 mM DTT and 5% sucrose) and sucrose buffer containing 20% sucrose and 4% paraformaldehyde using a Gradient Master (SKB). The samples were applied to the top of each gradient and centrifuged at 124,700$g$ at 4 °C for 16 h using a Beckman Coulter SW41Ti rotor. The fractions containing the DEK–nucleosome complexes were collected, desalted on a PD-10 column (Cytiva) and eluted in cryo-EM buffer (20 mM Tris-HCl pH 7.5 and 1 mM DTT). The resulting DEK–nucleosome sample was concentrated by using an Amicon Ultra 30-kDa filter (Millipore).

The preparation of the H1–nucleosome complex was performed as described above, with modifications. The 193-bp 601L nucleosome (1.25 µM) and H1 (4.34 µM) were mixed in reaction buffer (20 mM Tris-HCl pH 7.5, 0.25 mM DTT, 8.8% glycerol, 75 mM NaCl and 1.5 mM 2-mercaptoethanol). The gradient buffer was prepared using sucrose buffer (20 mM HEPES–KOH pH 7.5, 50 mM potassium acetate, 0.2 µM zinc acetate, 0.1 mM TCEP–HCl pH 7.0 and 10% sucrose) and sucrose buffer containing 25% sucrose and 0.1% glutaraldehyde. The sample was applied to the top of the gradient and centrifuged. The fractions containing the H1–nucleosome complexes were collected, desalted, eluted in buffer (20 mM HEPES–NaOH pH 7.5, 0.2 µM zinc acetate and 0.1 mM TCEP–HCl pH 7.0) and concentrated.

The preparation of the free nucleosome complex was performed as described above, with modifications. The gradient buffer was prepared using sucrose buffer (10 mM HEPES–NaOH pH 7.5, 20 mM NaCl, 1 mM DTT and 5% sucrose) and sucrose buffer containing 20% sucrose and 4% paraformaldehyde. The 193-bp 601L nucleosome (1.26 nmol) was applied to the top of the gradient and centrifuged. The fractions containing the nucleosome were collected, desalted, eluted in buffer (20 mM Tris-HCl pH 7.5 and 1 mM DTT) and concentrated. Finally, NP-40 was added to the sample to a final concentration of 0.00358%.

A 2.5-µl portion of each sample was applied to Quantifoil Cu R1.2/1.3 200-mesh grids, which were glow-discharged for 1 min by a PIB-10 Ion Bombarder (Vacuum Device). The grids were plunge-frozen using a Vitrobot Mark IV (Thermo Fisher Scientific) at 16 °C and 100% humidity.

## Cryo-EM analyses

For the DEK–nucleosome complexes, cryo-EM images were collected on a Krios G3i cryo-EM instrument (Thermo Fisher Scientific) equipped with a K3 BioQuantum imaging filter (Gatan), operated at 300 kV with a pixel size of 1.05 Å. In total, 8,110 videos of the DEK–nucleosome complex were recorded with a slit width of 25 eV, using the SerialEM software[60]. Each video was fractionated to 40 frames with a total dose of 55 $e^-$ per Å$^2$. The following processes were performed using RELION 3 and 4 (refs. [61,62]). Video frames were aligned and dose-weighted with MotionCor2 (ref. [63]). The contrast transfer function (CTF) was estimated using Gctf[64]. Particles were automatically picked with a box size of 210 × 210 pixels using the autopick module with two-dimensional (2D) class averages generated from reference-free autopicked particles with a Laplacian of Gaussian filter as a reference. Bad particles were removed by 2D classification. The initial model was generated by stochastic gradient descent (SGD)-based initial model, followed by 3D classification. In this step, the particles of the DEK$^2$ and DEK$^1$ complexes were separated. For the structure of the DEK$^2$–nucleosome complex,

the class containing DEK$^2$ was subjected to CTF refinement and Bayesian polishing. The resulting particles were then further classified with a mask around the nucleosomal linker DNA regions containing the two DEK molecules. The class containing the resolved DEK$^2$ structure was selected and refined. The resulting map was postprocessed with a mask covering the overall structure. The resolution of the final map was 2.8 Å, as estimated by the gold-standard Fourier shell correlation (FSC) = 0.143. For the structure of the DEK$^1$–nucleosome complex, the class containing DEK$^1$ was selected and particles were subjected to CTF refinement and Bayesian polishing. The resulting map was postprocessed with a mask covering the overall structure. The resolution of the final map was 3.2 Å, as estimated by gold-standard FSC = 0.143. The local resolutions were estimated using RELION.

For the H1–nucleosome complex, cryo-EM images were collected on a Krios G3i cryo-EM instrument (Thermo Fisher Scientific) equipped with a K3 BioQuantum image filter (Gatan), operated at 300 kV with a pixel size of 1.05 Å. In total, 8,340 videos of the H1–nucleosome complex were recorded with a slit width of 25 eV, using EPU software (Thermo Fisher Scientific). Each video was fractionated to 40 frames with a total dose of 56 $e^-$ per Å$^2$. Video frames were aligned and dose-weighted with MotionCor2 (ref. [63]). The CTF was estimated by Gctf[64]. Particles were automatically picked with a box size of 200 × 200 pixels using the autopick module in RELION 3.1, using 2D class averages generated from reference-free autopicked particles with a Laplacian of Gaussian filter as a reference. Bad particles were removed by 2D classification using RELION 3.1 (ref. [61]). The initial model was generated using SGD-based initial model, followed by 3D classification. The class containing H1 was selected and particles were subjected to CTF refinement and Bayesian polishing. The resulting particles were then further classified with a mask around the nucleosomal linker DNA regions containing H1. The resulting map was postprocessed with a mask covering the overall structure. The resolution of the final map was 4.0 Å, as estimated by gold-standard FSC = 0.143. The local resolution was estimated using RELION.

For the free nucleosome complex, cryo-EM images were collected on a Krios G4 cryo-EM instrument (Thermo Fisher Scientific) equipped with a K3 BioQuantum image filter (Gatan), operated at 300 kV with a pixel size of 1.06 Å. In total, 10,142 videos of the free nucleosome complex were recorded with a slit width of 20 eV, using EPU software (Thermo Fisher Scientific). Each video was fractionated to 40 frames with a total dose of 57 $e^-$ per Å$^2$. Video frames were aligned and dose-weighted with MotionCor2 (ref. [63]). The CTF was estimated by CTFFIND. Particles were picked automatically with a box size of 250 × 250 pixels using the autopick module in RELION 3.1, using 2D class averages generated from reference-free autopicked particles with a Laplacian of Gaussian filter as a reference; bad particles were removed by 2D classification using RELION 3.1 (ref. [61]). The initial model was generated using SGD-based initial model, followed by 3D classification. The classes containing nucleosomes were selected and particles were subjected to 3D autorefinement. The resulting particles were then further classified. On the basis of the linker DNA orientations, three nucleosome classes were selected and particles were subjected to CTF refinement and Bayesian polishing. The resulting maps were postprocessed with a mask covering each overall structure. The resolution of each final map was 3.4 Å (closed), 3.7 Å (middle) and 3.9 Å (open), as estimated by gold-standard FSC = 0.143. The local resolution was estimated using RELION.

## Model building

The atomic models of the DEK–nucleosome complexes were built on the basis of the crystal structure of the *Xenopus laevis* nucleosome containing the 197-bp palindromic Widom 601 positioning sequence (Protein Data Bank (PDB) 5LN0)[44]. The DEK structure was based on the AlphaFold2 model and manually inspected against the previously reported structure of the SAP domain (PDB 2JX3)[22]. The structures of

the nucleosome and the SAP domain were fitted by Chimera[65], edited manually using Coot[66] and ISOLDE[67] in ChimeraX[68] and refined against the cryo-EM maps using phenix.real_space_refine[69]. The amino acid residues of the *X. laevis* histones were adjusted to those of the human histones. The final structure was evaluated with MolProbity[70] equipped in the PHENIX package[71] (Table 1).

The atomic model of the H1–nucleosome complex was built on the basis of the crystal structure of the *X. laevis* nucleosome containing the 197-bp palindromic Widom 601 positioning sequence (PDB 5LN0) and the cryo-EM structure of the H1.2–nucleosome–RNA polymerase II complex (PDB 8HOV)[44,72]. The structures of the nucleosome and the H1 globular domain were fitted with Chimera[65], edited manually using Coot[66] and ISOLDE[67] and refined against the cryo-EM map using phenix.real_space_refine[69]. The amino acid residues of the *X. laevis* histones were adjusted to those of the human histones. The final structure was evaluated with MolProbity[70] equipped in the PHENIX package[71] (Table 1).

All structural figures were prepared using UCSF Chimera[65] and PyMOL (Schrödinger; http://www.pymol.org).

## Preparation of polynucleosome complexes for AFM imaging

The polynucleosome (0.1 µM calculated as a mononucleosome) was mixed with DEK (0.6 µM) in reaction buffer (3.2 mM Tris-HCl pH 7.5, 0.13 mM DTT, 1.1% glycerol, 7.5 mM NaCl, 0.015 mM EDTA, 0.003 mM PMSF and 0.06 mM 2-mercaptoethanol) and incubated at 37 °C for 30 min. The samples were dialyzed against buffer (10 mM Tris-HCl pH 7.5 and 1 mM DTT) at 4 °C. To confirm the DEK binding, the samples were subjected to agarose gel electrophoresis with ethidium bromide staining. In addition, to cleave the polynucleosomes into mononucleosomes, 70-ng portions of the samples were mixed with 10 U of ScaI (Takara) in reaction buffer (17 mM Tris-HCl pH 7.5, 0.7 mM DTT, 50 mM NaCl, 0.5 mM MgCl$_2$ and 0.1 mg ml$^{-1}$ BSA) and incubated at 22 °C for 12 h. The resulting samples were analyzed by nondenaturing PAGE with ethidium bromide staining.

## AFM imaging and analyses

A 20-µl aliquot of poly(L-ornithine) solution (0.01 mg ml$^{-1}$) (Wako) was applied onto freshly cleaved muscovite mica (Alliance Biosystems, 01873-CA) and incubated at room temperature for 2 min. The mica was washed twice with 1 ml of distilled H$_2$O. Afterward, 200 µl of AFM imaging buffer (10 mM Tris-HCl pH 7.5, 1 mM DTT and 0.03% NP-40) was applied to the mica and immediately removed. The polynucleosome samples (20 µl) were applied to the mica and incubated at room temperature for 5 min. The polynucleosome samples were washed twice with 200 µl of AFM imaging buffer and then 200 µl of AFM imaging buffer was deposited onto the mica. The AFM imaging was performed using a NanoWizard IIR instrument (JPK) with a BL-AC40TS-C2 Bio Lever Mini cantilever (Olympus, BL-AC40TS-C2). Images were acquired in the QI mode.

For the AFM image analyses, backgrounds and scan line artifacts were corrected using the Gwyddion software (David Nečas, Open Physics, 2012) and the area of each polynucleosome was measured by ImageJ software[73]. Statistical data analysis and graphing were performed by RStudio (https://posit.co/). The *P* values for comparing the polynucleosome areas were determined using the Mann–Whitney *U*-test.

## Histone methylation assay

DEK (0, 0.15, 0.3 and 0.6 µM) was mixed with the polynucleosome (0.4 µM calculated as a mononucleosome) containing the 208-bp repeat of the Widom 601 sequence or the mononucleosome (0.4 µM) containing the 208-bp Widom 601 sequence in the presence of the recombinant PRC2 core enzyme (0.15 µM) (Active Motif, 31887) and incubated for 3 h at 30 °C in 10 µl of reaction buffer (26 mM Tris-HCl pH 8.5, 0.3 mM DTT, 3% glycerol, 85 mM NaCl, 0.05 mM EDTA, 0.01 mM PMSF, 0.2 mM 2-mercaptoethanol, 5 mM HEPES–NaOH pH 7.5, 0.008%

Triton X-100, 0.14 mM TCEP and 0.1 mM *S*-adenosyl methionine). The resulting reaction mixtures were analyzed by western blotting or with an MTase-Glo methyltransferase assay kit (Promega).

For western blotting in Fig. 2b, the reaction mixtures were subjected to SDS–PAGE, transferred to PVDF membranes (0.2 µm) and blocked in Blocking One-P (Nacalai Tesque). The membranes were washed with TBS-T (20 mM Tris-HCl pH 7.5, 137 mM NaCl and 0.1% Tween-20) three times and then incubated overnight with the anti-H3K27me3 antibody (Cell Signaling, 9733, 1:1,000 dilution or 1E7, 1:1,000 dilution) in Can Get Signal immunoreaction enhancer solution 1 (Toyobo). The membranes were washed three times with TBS-T and then incubated with the secondary antibody fragment conjugated with horseradish peroxidase (HRP), anti-rabbit IgG, HRP-linked F(ab′)2 fragment donkey (Cytiva, NA9340; 1:3,000 dilution) or anti-mouse IgG HRP-linked F(ab′)2 fragment (Cytiva, NA9310; 1:5,000 dilution) in Can Get Signal immunoreaction enhancer solution 2 (Toyobo). After three washes with TBS-T, the chemical luminescence signal was detected using enhanced chemiluminescence prime western blotting detection reagents (Cytiva) by an Amersham Imager 680. For Extended Data Fig. 4d, the reaction mixtures were subjected to SDS–PAGE, transferred to nitrocellulose membranes (Amersham Protran, 0.1 µm, 10600000) and blocked in Blocking One-P (Nacalai Tesque). The membranes were washed with TBS-T three times and then incubated overnight with the rabbit anti-H3K27me3 antibody (Cell Signaling, 9733; 1:1,000 dilution) and the mouse anti-H2B antibody (Cell Signaling, 2934; 1:1,000 dilution) in Can Get Signal immunoreaction enhancer solution 1 (Toyobo). The membranes were washed three times with TBS-T and then incubated with goat anti-rabbit IgG conjugated with Alexa Fluor 488 (Jackson Immuno Research Laboratories, 111-545-144; 1:500 dilution) and goat anti-mouse IgG conjugated with Alexa Fluor 647 (Invitrogen, A32728; 1:5,000 dilution) in Can Get Signal immunoreaction enhancer solution 2 (Toyobo). After four washes with TBS-T, the signals were detected with an Amersham Typhoon imager (Cytiva).

To quantify the PRC2 methyltransferase activity, the reaction mixtures were treated according to the MTase-Glo methyltransferase assay kit manual (Promega). The resulting samples were mixed with 2 µl of 6× MTase-Glo reagent and incubated at room temperature for 30 min. Then, 12 µl of the MTase-Glo detection solution was added to the samples. After incubation for 30 min at room temperature, 20 µl of the reaction mixture was transferred to a 384-well plate (Corning, 4513). The luminescence of the samples was measured using a CLARIOstar Plus plate reader (BMG Labtech).

## Preparation of PRC2 protein for the pulldown assay

The DNA fragments encoding human EZH2, SUZ12, EED and RBBP4 were inserted into the pCMV vector. EZH2 was fused with a His$_6$ tag at the N terminus and EED was fused with a FLAG tag at the N terminus. These proteins were expressed in Expi293F cells (Gibco). The cells were diluted to a final density of $3.0 \times 10^6$ cells per ml in 1 l of Expi293 expression medium (Gibco). Then, 1 ml of pCMV DNA (1 mg ml$^{-1}$) was mixed with 120 ml of OPTI-MEM 1 (Gibco) and 3 ml of PEI MAX (1 mg ml$^{-1}$) with transfection-grade linear polyethylenimine hydrochloride (molecular weight: 40,000; Polysciences) and incubated at room temperature for 15 min. The cells were transfected with the DNA complexes and cultured at 37 °C in an 8% CO$_2$ atmosphere for 2 days. The cells were harvested by centrifugation, washed with PBS, flash-frozen in liquid nitrogen and stored at −80 °C. The cells were resuspended in Expi293 lysis buffer (50 mM Tris-HCl pH 8.0, 250 mM NaCl, 1% NP-40, 1 mM PMSF and 0.05 mg ml$^{-1}$ DNase I; Wako) and disrupted by sonication. After centrifugation, the supernatant was subjected to FLAG tag affinity chromatography using anti-FLAG M2 affinity gel (Sigma-Aldrich) and eluted with elution buffer (50 mM Tris-HCl pH 8.0, 250 mM NaCl, 1 mM TCEP, 1 mM PMSF and 0.3 mg ml$^{-1}$ Flag peptide). The eluted sample was purified by Superose 6 Increase HiScale 16/40 (Cytiva) size-exclusion column chromatography with a buffer comprising 50 mM Tris-HCl

pH 7.5, 300 mM NaCl, 1 mM TCEP and 5% glycerol. The purified sample was concentrated by using an Amicon Ultra 30-kDa filter (Millipore), dialyzed against PRC2 storage buffer (25 mM Tris-HCl pH 7.5, 150 mM NaCl, 10% glycerol and 1 mM TCEP), frozen in liquid nitrogen and stored at −80 °C.

## PRC2 pulldown assay

DEK (0 and 1 μM) was mixed with the His–PRC2 complex (0 and 0.5 μM) in 20 μl of reaction buffer (22.5 mM Tris-HCl pH 7.5, 0.5 mM DTT, 10% glycerol, 150 mM NaCl, 0.25 mM EDTA, 0.05 mM PMSF and 0.5 mM TCEP). The samples were mixed with 5 μl of Ni-NTA resin (Qiagen) and 30 μl of pulldown buffer (20 mM Tris-HCl pH 8.0, 150 mM NaCl, 0.1% NP-40, 5% glycerol and 20 mM imidazole) and incubated at 4 °C for 1 h. After collection of supernatants, the beads were washed twice with 500 μl of pulldown buffer. The resulting samples were analyzed by SDS–PAGE and visualized by Oriole fluorescent gel stain (Bio-Rad).

## Mouse maintenance and preparation of pregnant mice

JCL:ICR (CLEA) and Slc:ICR (Kapan) mice were housed in a temperature-controlled and humidity-controlled environment (23 ± 3 °C and 50% ± 15%, respectively) under a 12-h light–dark cycle. Animals were housed in sterile cages (Innocage, Innovive) containing bedding chips (PALSOFT, Oriental Yeast), with 2–6 mice per cage, and provided irradiated food (CE-2, CLEA) and filtered water ad libitum.

## ChIP-seq

Two replicates of approximately $10^7$ cells dissociated from the telencephalon (Neuron Dissociation Solution; Fujifilm Wako) of E11.5 mouse embryos or approximately $10^7$ HEK293 cells were subjected to ChIP-seq[25]. The HEK293 cells were authenticated on the basis of their morphology. Cells were fixed with 1% formaldehyde and stored at −80 °C until analysis. The thawed cells were suspended in radioimmunoprecipitation assay (RIPA) buffer for sonication (10 mM Tris-HCl pH 8.0, 1 mM EDTA, 140 mM NaCl, 1% Triton X-100, 0.1% SDS and 0.1% sodium deoxycholate) and sonicated using a Picoruptor (15 cycles of 30-s on and 30-s off; Diagenode). The cell lysates were diluted with RIPA buffer for immunoprecipitation (50 mM Tris-HCl pH 8.0, 150 mM NaCl, 2 mM EDTA, 1% Nonidet P-40, 0.1% SDS and 0.5% sodium deoxycholate), incubated for 1 h at 4 °C with protein A/G magnetic beads (Pierce) to clear nonspecific reactivity and then incubated overnight at 4 °C with protein A/G magnetic beads (Pierce), which were preincubated with an anti-DEK antibody (BD, 610948) overnight at 4 °C. The isolated beads were washed three times with wash buffer (2 mM EDTA, 150 mM NaCl, 0.1% SDS, 1% Triton X-100 and 20 mM Tris-HCl pH 8.0) and then once with wash buffer containing 500 mM NaCl. Immunoprecipitated chromatin complexes were eluted from the beads for 15 min at 65 °C in buffer (10 mM Tris-HCl, pH 8.0, 5 mM EDTA, 300 mM NaCl and 0.5% SDS) and then subjected to protein digestion with proteinase K (Nacalai) for >6 h at 37 °C. The crosslinks were removed by incubation at 65 °C for >6 h and the remaining DNA was purified by phenol–chloroform–isoamyl alcohol extraction and ethanol precipitation. The DNA was washed with 70% ethanol and suspended in water. The sequence library was prepared from the immunoprecipitated DNA with an NEBNext Ultra II FS DNA library prep kit for Illumina (New England Biolabs) and subjected to deep sequencing on the HiseqX platform (Illumina) to yield 151-bp paired-end reads. Approximately 20 million sequences were obtained from each sample. For H3K27me3 ChIP-seq using HEK293 cells, we obtained raw sequence files from the Gene Expression Omnibus (GEO) database (GSE133391 and GSM3907592[74]). A total of 101 bp at the 3′ end were trimmed using the FASTX-Toolkit's FASTX trimmer function and the adaptor sequence was removed using the Cutadapt software. The reads were subsequently mapped to a mouse genome (mm10) or a human genome (hg38) using Bowtie2 (ref. 75). Among the uniquely mapped reads, those in the blacklisted regions determined by the Encyclopedia of DNA Elements project[76,77]

and sex chromosomes were removed using BEDTools[78] and duplicated reads were removed using the SAMtools rmdup function[79].

Genes annotated in GENCODE vM20 were used to determine the genome features. Promoters were defined as the upstream region 1 kbp from the TSSs and gene bodies included exons and introns. Other regions were determined as those that were not characterized. The peaks were determined by the f-seq software[80]. The genes associated with the peaks were identified using the BEDTools annotation software.

## Cleavage under targets and tagmentation (CUT&Tag)

Two biological replicates of $10^4$ NPCs, isolated as CD133-high cells from the neocortex of E11.5 mouse embryos, were subjected to H3K27me3 or H2AK119ub1 CUT&Tag[25,81,82]. Concanavalin A beads (Bangs Laboratories, BP531) were mixed with NPCs, washed with wash buffer (150 mM NaCl, 0.5 mM spermidine, protease inhibitor (Roche) and 20 mM HEPES, pH 7.5) and incubated in primary antibody buffer (150 mM NaCl, 0.5 mM spermidine, protease inhibitor, 0.05% digitonin, 2 mM EDTA, 0.1% BSA and 20 mM HEPES, pH 7.5) with an anti-H3K27me3 antibody (Cell Signaling, 9733S; 1:100 dilution) or an anti-H2AK119ub1 antibody (Cell Signaling, 8240S; 1:100 dilution) for 16 h at 4 °C. After washing with Dig-wash buffer (150 mM NaCl, 0.5 mM spermidine, protease inhibitor, 0.05% digitonin and 20 mM HEPES, pH 7.5), the beads were incubated with the secondary antibody (Rockland, 611-201-122) in Dig-wash buffer for 30 min at 24 °C, washed with Dig-wash buffer, incubated with the pAG-Tn5 adaptor complex for 1 h at 24 °C, washed with Dig-300 buffer (300 mM NaCl, 0.5 mM spermidine, protease inhibitor, 0.01% digitonin and 20 mM HEPES, pH 7.5) and incubated with Dig-300 buffer containing 10 mM $MgCl_2$ for 1 h at 37 °C. The reaction was quenched with 17 mM EDTA and the DNA was eluted with 0.1% SDS and decrosslinked for 16 h at 65 °C. After treatment with 0.17 mg ml⁻¹ proteinase K for 30 min at 50 °C, the DNA was purified by phenol–chloroform–isoamyl alcohol extraction and ethanol precipitation, washed with 70% ethanol, suspended in water and amplified by PCR using Q5 Hot Start high-fidelity 2× master mix (New England Biolabs) as follows: 5 min at 72 °C and 30 s at 98 °C, followed by 12 cycles of 10 s at 98 °C and 20 s at 63 °C, with a final extension at 72 °C for 1 min. The amplified DNA was purified using solid-phase reversible immobilization (SPRI) beads (Beckman Coulter). Deep sequencing and subsequent analysis were performed as described for ChIP-seq.

## ATAC-seq

Four biological replicates of $10^4$ NPCs, isolated as CD133-high cells from the neocortex of E11.5 mouse embryos, were subjected to ATAC-seq[83]. NPCs were suspended in ATAC lysis buffer (10 mM Tris-HCl pH 7.4, 10 mM NaCl, 3 mM $MgCl_2$ and 0.1% NP-40), centrifuged at 1,000g for 5 min at 4 °C and tagmentated with Tagmentase (Illumina, 1:20 dilution) in Tagmentation buffer (Illumina) at 37 °C for 30 min. Tagmentated DNA was purified with a DNA purification kit (Nippon Genetics, FG-91302) and amplified by PCR with Q5 Hot Start high-fidelity 2× master mix (New England Biolabs, M0494L) as follows: 5 min at 72 °C and 30 s at 98 °C, followed by 12 cycles of 10 s at 98 °C and 20 s at 63 °C, with a final extension at 72 °C for 1 min. The amplified DNA was purified using SPRI beads (Beckman Coulter, B23318). The libraries were subjected to deep sequencing using the DNB-seq platform (MGI) to yield 100-bp paired-end reads and subsequent analyses were performed as described for ChIP-seq.

## RNA-seq

RNA-seq libraries were prepared using three biological replicates of NPCs, isolated as CD133-high cells from the neocortex of E11.5 mouse embryos with a SMART-seq stranded kit (Takara) and subjected to deep sequencing on a HiseqX platform (Illumina) to yield 151-bp paired-end reads. Approximately 20 million sequences were obtained from each sample. Then, 101 bp at the 3′ end were trimmed using the FASTX-Toolkit's FASTX trimmer function and the adaptor sequence was

removed using the Cutadapt software. For RNA-seq using HEK293 cells, we obtained raw sequence files from the GEO database (GSE249290, GSM7933161, GSM7933162 and GSM7933163)[84]. The reads were subsequently mapped to a mouse genome (mm10) or a human genome (hg38) using hisat2 (ref. 85). Among the uniquely mapped reads, those in the blacklisted regions determined by the Encyclopedia of DNA Elements project[76,77] and sex chromosomes were removed using BEDTools[78]. The fragments per kilobase of exon per million mapped reads (FPKM) were calculated as the expression level value.

Genes with high, medium and low expression were determined as follows: high, FPKM > 10; medium, 1 < FPKM < 10; low, 1 > FPKM > 0.1.

### Generation of *Dek*-KO mice and genotyping PCR

CRISPR–Cas9-mediated genome editing was performed using the iGONAD method[86]. First, 2–3-month-old female mice (Jcl:ICR, CLEA) were mated with male mice the day before electroporation. The female mice with vaginal plugs were used for iGONAD at E0.75. Genome-editing solution was prepared with 1 mg ml$^{-1}$ Cas9 protein (Integrated DNA Technologies (IDT), 1081059), 30 mM CRISPR RNA (crRNA; annealed with *trans*-activating crRNA; IDT, 1072534) and FastGreen (Fujifilm Wako, 061-00031) in OPTI-MEM (Thermo Fisher Scientific, 11058021). The exposed oviducts of the female mice were injected wih the solution through microcapillary injection. Oviduct electroporation was performed using NEPA21 and CUY652P2.5×4 (NEPA Gene) with the following protocol: three poring pulses (50 V, 5 ms, 50-ms interval and 10% decay; ±pulse orientation) and three transfer pulses (10 V, 50 ms, 50-ms interval and 40% decay; ±pulse orientation). After electroporation, the oviducts were returned to their original position. The crRNA and single-stranded oligodeoxynucleotide (ssODN) sequences were as follows: crRNA for upstream, 5′-GCCACCACGTTTGTTTCCGA-3′; crRNA for downstream, 5′-ATGCTGACCACTTACTATTA-3′.

For genotyping, a mouse finger was amputated and lysed according to the quick DNA purification protocol provided by The Jackson Laboratory (https://www.jax.org/jax-mice-and-services/customer-support/technical-support/genotyping-resources/dna-isolation-protocols). PCR was performed with QuickTaq (Toyobo) using the following three PCR primers: 5′-ACGGGAAGCCTCGCAAGTAG-3′, 5′-GCTGCCTGATAAAGTGACACTAG-3′ and 5′-AGCATGTGTGAAACGGTAAGTG-3′. The amplicon was analyzed by agarose gel electrophoresis.

### Western blotting

For the western blotting in Extended Data Fig. 10, the neocortexes from E11.5 embryos were lysed with sample buffer (Nacalai Tesque) and fractionated by SDS–PAGE (precast gel; Bio-Rad, 4561096). Proteins were transferred to the membrane (nitrocellulose transfer packs; Bio-Rad, 1704159), which was blocked in blocking reagent for Can Get Signal (Toyobo). The membranes were washed with TBS-T and then incubated overnight with the anti-DEK (BD, 610948; 1:500 dilution), anti-GAPDH (Cell Signaling, 2118S; 1:500 dilution), anti-H3K27me3 (Cell Signaling, 9733; 1:500 dilution) or anti-H3 (Active Motif, 39763; 1:5,000 dilution) antibody in Can Get Signal immunoreaction enhancer solution 1 (Toyobo). The membranes were washed with TBS-T and then incubated with the secondary antibody fragment conjugated with anti-rabbit IgG, HRP-linked whole antibody donkey (Cytiva, NA934VS; 1:3,000 dilution) or anti-mouse IgG, HRP-linked whole antibody sheep (Cytiva, NA931VS; 1:5,000 dilution) in Can Get Signal immunoreaction enhancer solution 2 (Toyobo). After washing with TBS-T, the signals were detected using the Immobilon western chemiluminescent HRP substrate (Millipore) by an iBright imager (Thermo Fisher Scientific).

### Immunohistochemistry

Embryos at E11.5 fixed with 4% paraformaldehyde at 4 °C for 2 h were washed with PBS, incubated at 4 °C in 10%, 20% and 30% sucrose in PBS until they sank to the bottom of the tube, frozen in optimal cutting temperature compound (Sakura) and stored at −80 °C. Serial 14-μm coronal cryosections were sliced on a Cryostat (Tissue-Tek) and stored at −80 °C until use. The sections were washed in TBS-T and blocked in 3% BSA and 0.1% Triton X-100 in TBS. The sections were immunostained with anti-DEK (BD, 610948; 1:500 dilution) overnight at 4 °C, followed by staining with donkey anti-mouse Alexa Fluor 555 (Thermo Fisher Scientific; 1:1,000 dilution) and Hoechst (Thermo Fisher Scientific, H3570; 1:10,000 dilution) for 1 h at 24 °C. Primary antibodies, secondary antibodies and Hoechst were used in blocking buffer. Slides were mounted with mounting medium (Vectashield vibrance antifade mounting medium; Vector Laboratories, H-1700) before imaging. Images were acquired using an FV3000 confocal microscope (Olympus) and processed using Fiji[87].

### Reporting summary

Further information on research design is available in the Nature Portfolio Reporting Summary linked to this article.

## Data availability

The cryo-EM reconstructions and atomic models of the DEK$^2$–nucleosome, DEK$^1$–nucleosome and H1–nucleosome complexes were deposited to the EM Data Bank and the PDB under the following accession codes: DEK$^2$–nucleosome, EMD-37115 and PDB 8KCY; DEK$^1$–nucleosome, EMD-37121 and PDB 8KD1; H1–nucleosome, EMD-37149 and PDB 8KE0; free nucleosome (closed), EMD-38443; free nucleosome (middle), EMD-38444; free nucleosome (open), EMD-38445. The sequence data were deposited in the DNA Data Bank of Japan Sequence Read Archive under the following accession codes: DRA016775 (DEK ChIP-seq using mouse embryonic brain), DRA019123 (DEK ChIP-seq using HEK293 cells), DRA016561 (H3K27me3 ChIP-seq), DRA016776 (H3K27me3 CUT&Tag), DRA016778 (H2AK119ub1 CUT&Tag), DRA016781 (ATAC-seq) and DRA016780 (RNA-seq). We also used published datasets GSE133391 (H3K27me3 ChIP-seq using HEK293 cells) and GSE249290 (RNA-seq using HEK293 cells). The *Dek*-KO mouse is available from the corresponding authors upon reasonable request. The DEK SAP structure was downloaded from the PDB under accession code 2JX3. Source data are provided with this paper.

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

## Acknowledgements

We thank Y. Iikura, N. Uruma, Y. Takeda, H. Tanaka, M. Ogasawara and N. Kuwayama (University of Tokyo) for their assistance and M. Kikkawa (University of Tokyo) for cryo-EM data collection. We also thank A. Tsutsumi (University of Tokyo) and M. Wolf (Okinawa Institute of Technology) for their contribution in initial stage of the cryo-EM experiments and The University of Tokyo Institute for Quantitative Biosciences Olympus Bioimaging Center for microscopy. This work was supported in part by Japan Society for the Promotion of Science KAKENHI (JP20H03201, JP20H05690, JP22H15033, JP23K17392 and JP24H00062 to T.K.; JP23H05475 and JP24H02328 to H. Kurumizaka; JP22KJ0871 to K.E.; JP22K06098 to Y.T.; JP23H04214, JP24K02020 and JP16H06279 to Y.K.; JP24H02325 and JP21H04764 to H. Kimura; JP22K06183 to H.M.; JP22H00431 and JP22H04925 to Y.G.), Japan Science and Technology Agency (JST) Exploratory Research for Advanced Technology (JPMJER1901 to H. Kurumizaka), JST Core Research for Evolutional Science and Technology (CREST) (JPMJCR24T3 to H. Kurumizaka), Agency for Medical Research and Development CREST (JP23gm1310004 to Y.G.) and Research Support Project for Life Science and Drug Discovery (JP24ama121009 to H. Kurumizaka; JP24ama121002 to M. Kikkawa). The funders had no role in study design, data collection and analysis, decision to publish or preparation of the manuscript.

## Author contributions

Conceptualization, T.K., K.E., Y.K., Y.G. and H. Kurumizaka. Methodology, T.K., K.E., Y.K., M.S., T.I., L.N., H.M., Y.T., Y.G. and H. Kurumizaka. Resources, T.K., K.E., Y.K., T.I., J.K., H. Kimura, Y.T., Y.G. and H. Kurumizaka. Investigation, T.K., K.E., Y.K., M.S., T.I., L.N., H.M. and Y.T. Visualization, T.K., K.E., Y.K. and H. Kurumizaka. Funding acquisition, T.K., K.E., Y.K., H.M., H. Kimura, Y.T., Y.G. and H. Kurumizaka. Project administration, T.K. and H. Kurumizaka. Supervision, Y.G. and H. Kurumizaka. Writing—original draft, T.K., K.E., Y.K. and H. Kurumizaka. Writing—review and editing, T.K., K.E., Y.K., Y.G. and H. Kurumizaka.

## Competing interests

The authors declare no competing interests.

## Additional information

**Extended data** is available for this paper at https://doi.org/10.1038/s41594-025-01493-w.

**Correspondence and requests for materials** should be addressed to Yukiko Gotoh or Hitoshi Kurumizaka.

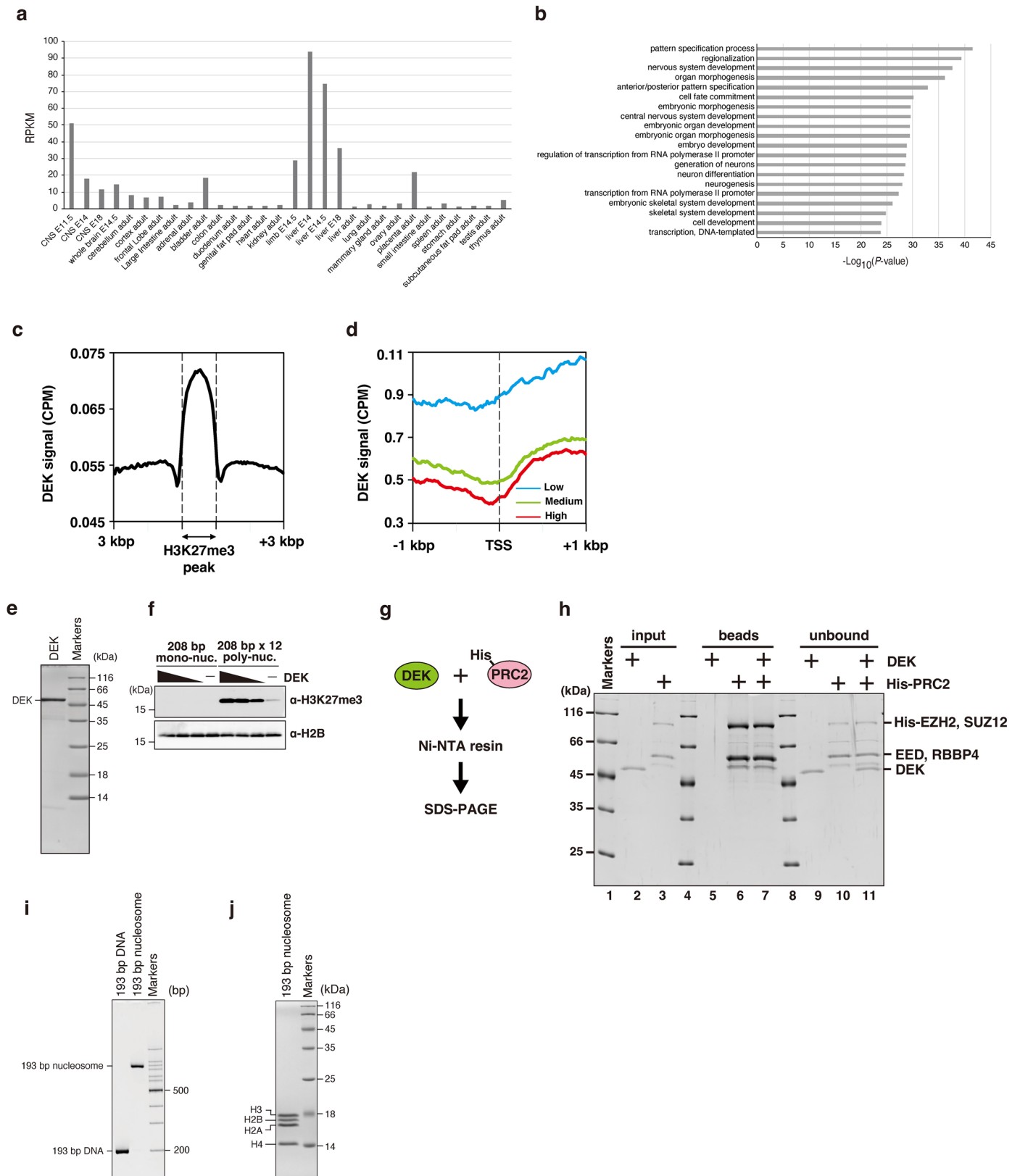

**Extended Data Fig. 1 | See next page for caption.**

**Extended Data Fig. 1 | DEK expression, localization, and H3K27me3 regulation of DEK.** (**a**) Expression profile of mouse DEK RNA in various tissues. RNA profiling data sets were generated by the Mouse ENCODE project PRJNA66167. (**b**) Gene ontology (GO) analysis of the DEK-bound genes. Top 20 GO terms of the genes are shown. (**c**) Localization of DEK in HEK293 cells. DEK signal around H3K27me3 peaks. (**d**) DEK localization around TSSs in genes with low, medium, or high expression. (**e**) The purified DEK protein was analyzed by SDS-PAGE with Coomassie Brilliant Blue (CBB) staining. This experiment is reproduced more than three times with consistent results. (**f**) H3K27me3 deposition assay of mono-nucleosomes and poly-nucleosomes. The mono-nucleosome contains the 208 base-pair Widom 601 DNA and the poly-nucleosome contains twelve repeats of the 208 base-pair Widom 601 DNA. The loading controls (H2B) are shown. This experiment is reproduced more than three times with consistent results. (**g**) Schematic illustration of the DEK and PRC2 pull-down assay. Recombinant DEK and the His$_6$-tagged PRC2 complex were mixed with Ni-NTA beads and analyzed by SDS-PAGE. His-PRC2 bound to the Ni-NTA beads. (**h**) SDS-PAGE of the resultant samples. This experiment was repeated twice and consistent results were obtained. (**i**) The purified 193 bp DNA and the 193 bp nucleosome were analyzed by nondenaturing-PAGE with ethidium bromide staining. This experiment is reproduced more than three times with consistent results. (**j**) The purified 193 bp nucleosome was analyzed by SDS-PAGE with CBB staining. This experiment is reproduced more than three times with consistent results.

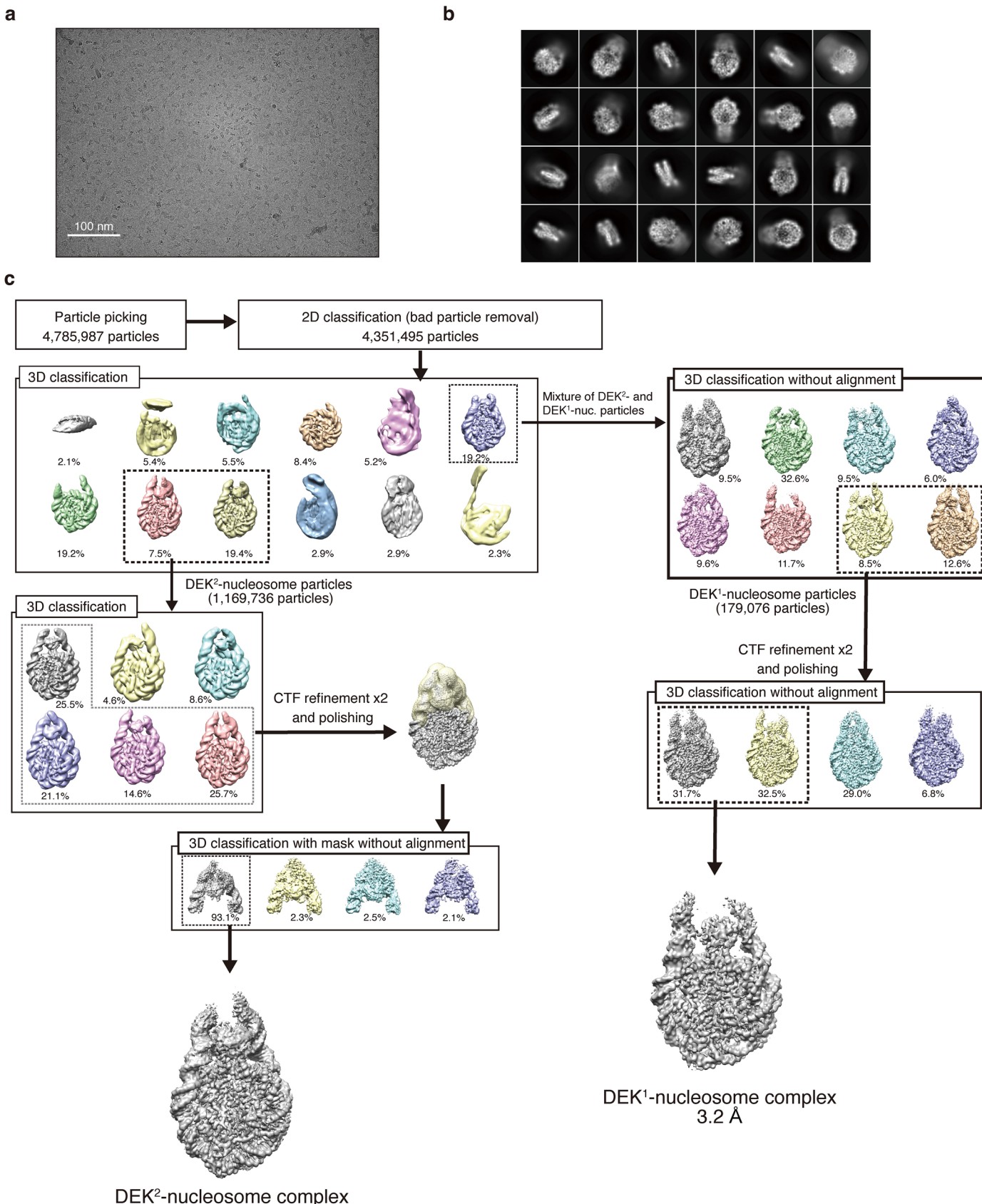

**Extended Data Fig. 2 | Cryo-EM data collection and image processing of the DEK-nucleosome complexes.** (**a**) Representative micrograph of the DEK-nucleosome complexes. 8110 micrographs were collected. (**b**) Representative 2D classes of the DEK-nucleosome complexes. (**c**) Workflow of the image processing.

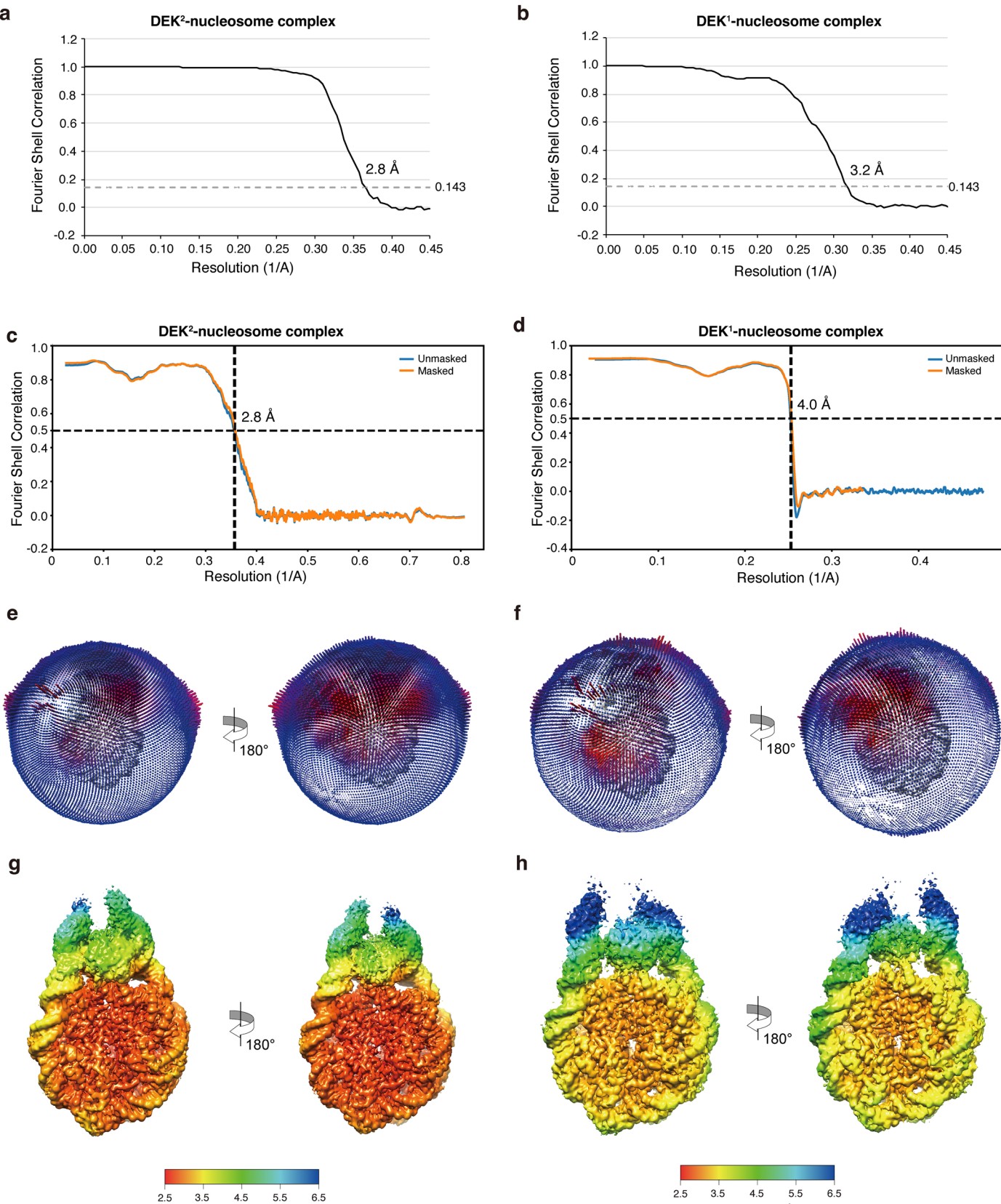

**Extended Data Fig. 3 | Quality of the DEK-nucleosome complexes.**
(**a,b**) Fourier Shell Correlation (FSC) curves for the DEK²-nucleosome (a) and
DEK¹-nucleosome (b) complexes. The resolutions of the structures of the
DEK²-nucleosome and DEK¹-nucleosome complexes were estimated to be
2.8 Å and 3.2 Å, respectively, by an FSC = 0.143. (**c,d**) Map to model FSC

curves for the DEK²-nucleosome (c) and DEK¹-nucleosome (d) complexes.
(**e,f**) Euler angular distributions of the structures of the DEK²-nucleosome (e) and
DEK¹-nucleosome (f) complexes. (**g,h**) Local resolution maps of the structures of
the DEK²-nucleosome (g) and DEK¹-nucleosome (h) complexes.

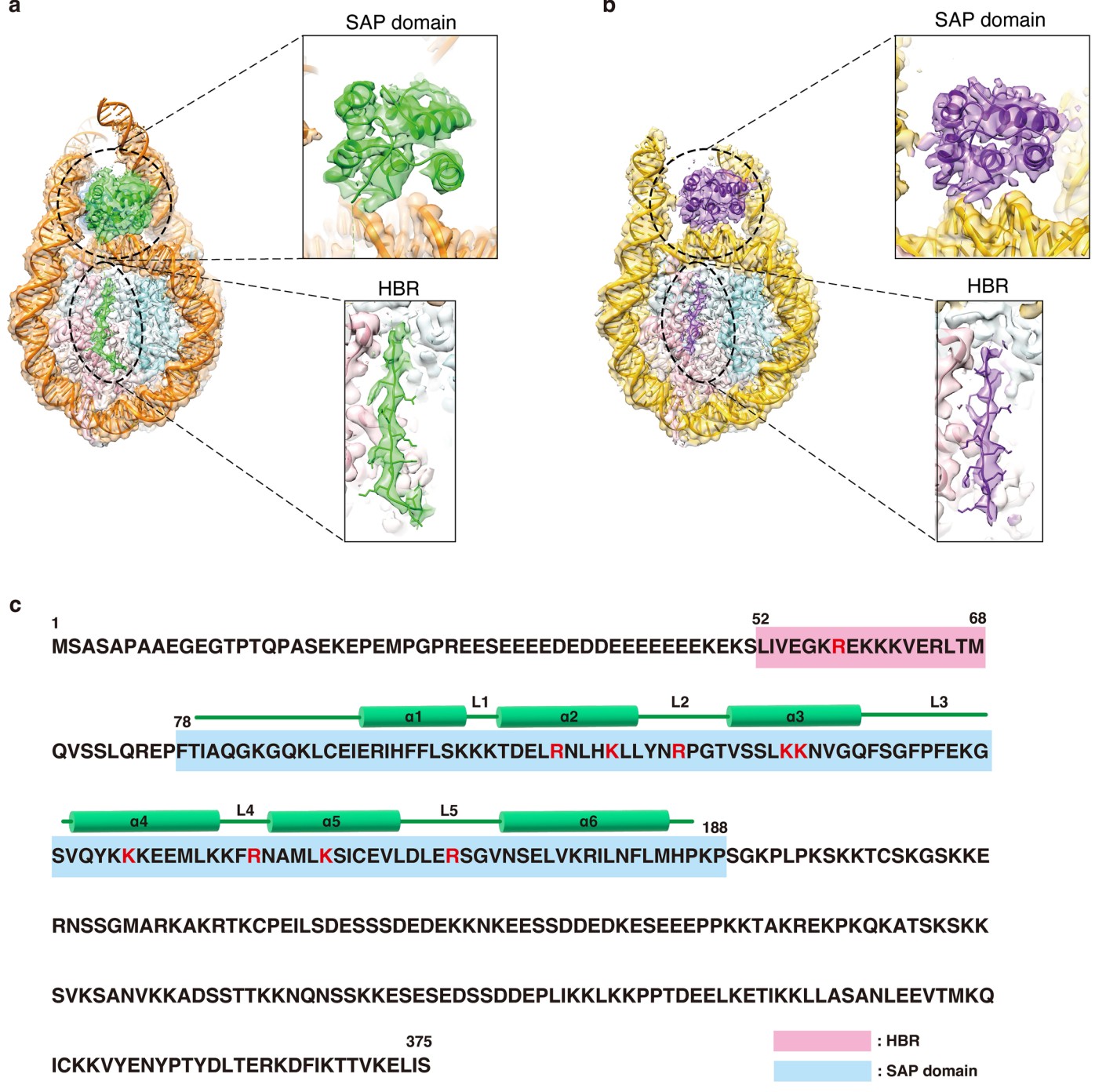

**Extended Data Fig. 4 | Cryo-EM density of the DEK-nucleosome complexes.** (**a**) Close-up views of the SAP domain and HBR models of the DEK dimer in the nucleosome fitted with the EM density map. (**b**) Close-up views of the SAP domain and HBR models of the DEK monomer in the nucleosome fitted with the EM density map. (**c**) The amino acid sequence of human DEK. The histone binding region (HBR) and SAP domain are highlighted in pink and blue, respectively. The residues involved in the DNA binding are shown with red characters. The α-helices of SAP domain are presented with green cylinders.

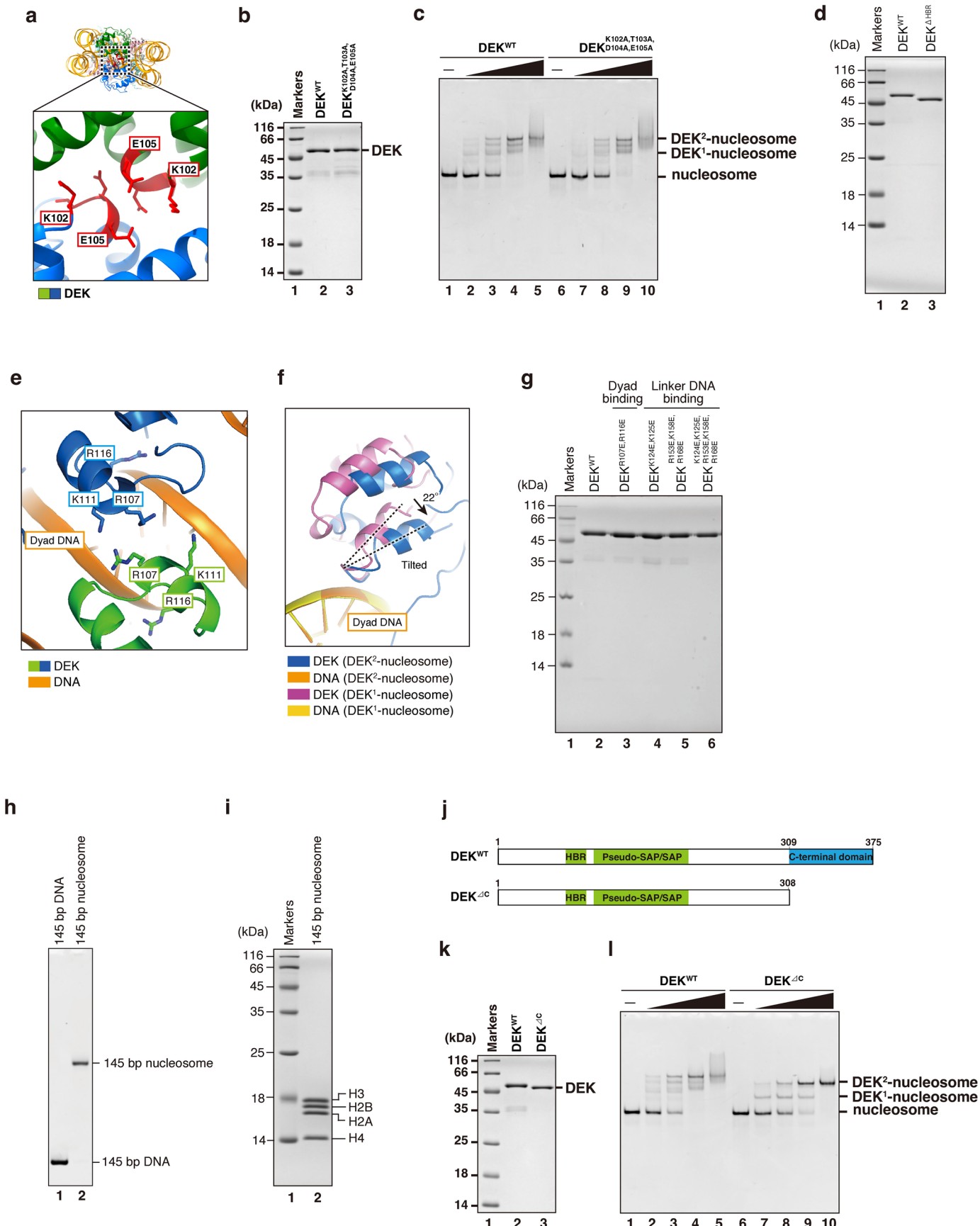

**Extended Data Fig. 5 | See next page for caption.**

**Extended Data Fig. 5 | The mutational analyses and spatial arrangement of DEK SAP domain.** (**a**) Close-up view of the atomic model of the DEK-DEK interface. Amino acids potentially responsible for the DEK-DEK interaction, K102 to E105, are highlighted in red. (**b**) The purified $DEK^{WT}$ and $DEK^{K102A,T103A,D104A,E105A}$ proteins were analyzed by SDS-PAGE. (**c**) Gel mobility shift assay of DEKWT (lanes 1-5) and $DEK^{K102A,T103A,D104A,E105A}$ (lanes 6-10) with the 193 bp nucleosome. The samples were analyzed by native-PAGE. The results were reproduced in three independent experiments. (**d**) The purified $DEK^{WT}$ and $DEK^{ΔHBR}$ proteins were analyzed by SDS-PAGE with CBB staining. (**e**) Close-up view of two DEK molecules binding on the dyad DNA in the nucleosome. (**f**) Structural comparison of the DEK molecules between the $DEK^2$-and $DEK^1$- nucleosome complexes.

The DEK molecule of the $DEK^2$-nucleosome is tilted by 22° from the $DEK^1$-nucleosome on the nucleosomal dyad DNA region. (**g**) The purified $DEK^{WT}$, $DEK^{R107E,R116E}$, $DEK^{K124E,K125E}$, $DEK^{R153E,K158E,R168E}$, and $DEK^{K124E,K125E,R153E,K158E,R168E}$ proteins were analyzed by SDS-PAGE with CBB staining. (**h**) The purified 145 bp DNA and the 145 bp nucleosome were analyzed by nondenaturing-PAGE with ethidium bromide staining. (**i**) The purified 145 bp nucleosome was analyzed by SDS-PAGE with CBB staining. (**j**) Schematic illustration of $DEK^{WT}$ and $DEK^{△C}$. (**k**) The purified $DEK^{WT}$ and $DEK^{△C}$ proteins were analyzed by SDS-PAGE. (**l**) Gel mobility shift assay of $DEK^{WT}$ (lanes 1-5) and $DEK^{△C}$ (lanes 6-10) with the 193 bp nucleosome. The samples were analyzed by native-PAGE. The results were reproduced in three independent experiments.

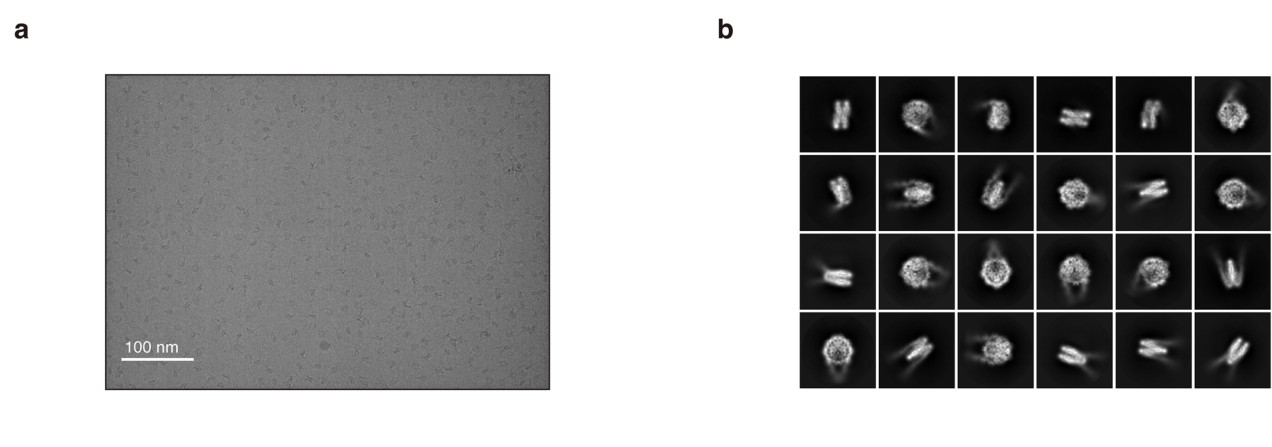

**Extended Data Fig. 6 | Cryo-EM data collection and image processing of the free-nucleosomes.** (**a**) Representative micrograph of the free-nucleosomes. 10142 micrographs were collected. (**b**) Representative 2D classes of the free-nucleosomes. (**c**) Workflow of the image processing.

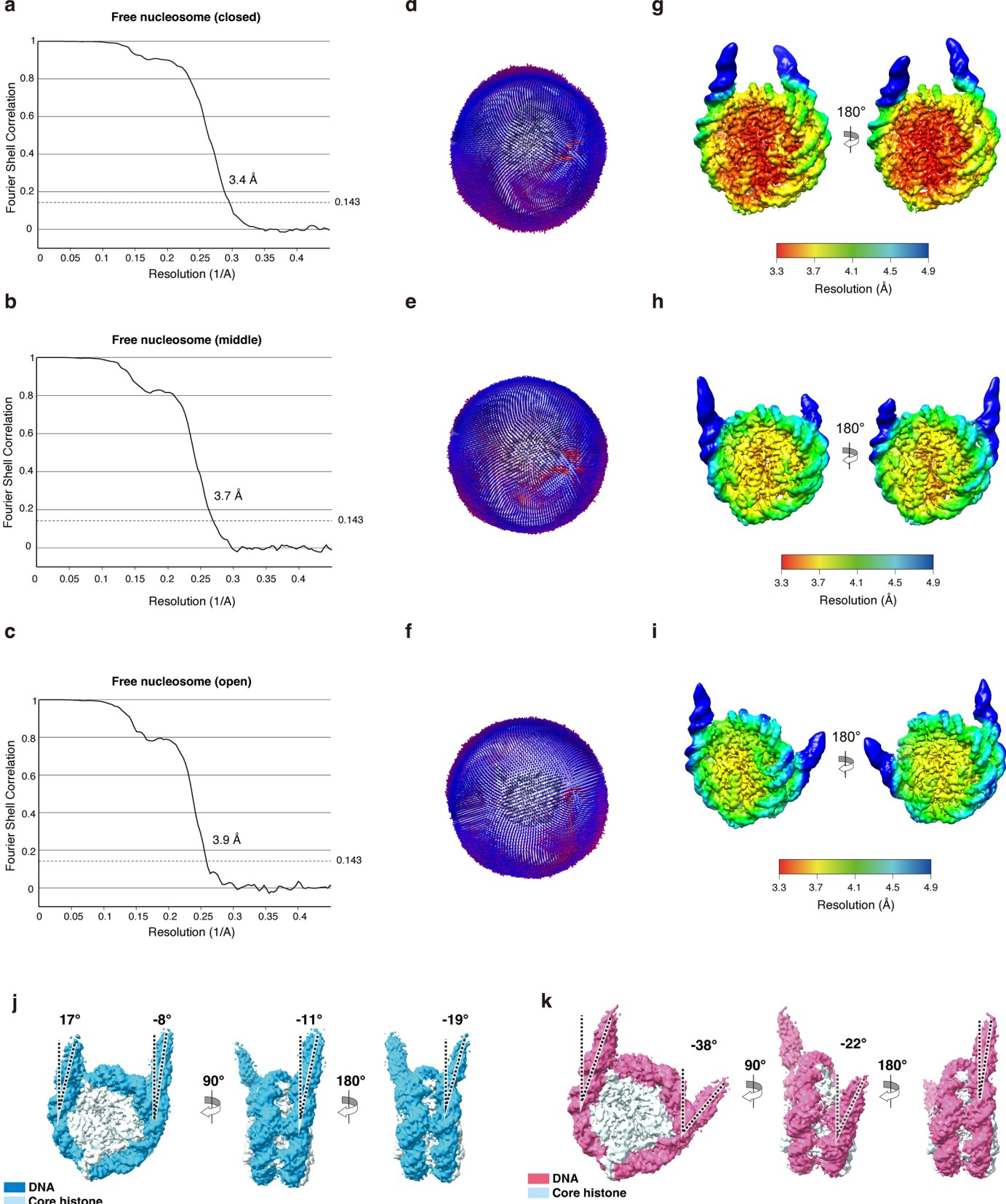

**Extended Data Fig. 7 | The free-nucleosome structures. (a-c)** Fourier Shell Correlation (FSC) curve for the free-nucleosome complex. The resolutions of the structures of the nucleosome$^{closed}$ (a), nucleosome$^{middle}$ (b), and nucleosome$^{open}$ (c) were estimated to be 3.4, 3.7, and 3.9 Å by an FSC = 0.143, respectively. **(d-f)** Euler angular distributions of the structures of the nucleosome$^{closed}$ (d),

nucleosome$^{middle}$ (e), and nucleosome$^{open}$ (f). **(g-i)** Local resolution maps of the structures of the nucleosome$^{closed}$ (g), nucleosome$^{middle}$ (h), and nucleosome$^{open}$ (i). **(j, k)** The cryo-EM maps of the nucleosome$^{middle}$ (j) and nucleosome$^{open}$ (k) are presented with the angles of the linker DNA in parallel and perpendicular directions.

**a**

**b**

**c**

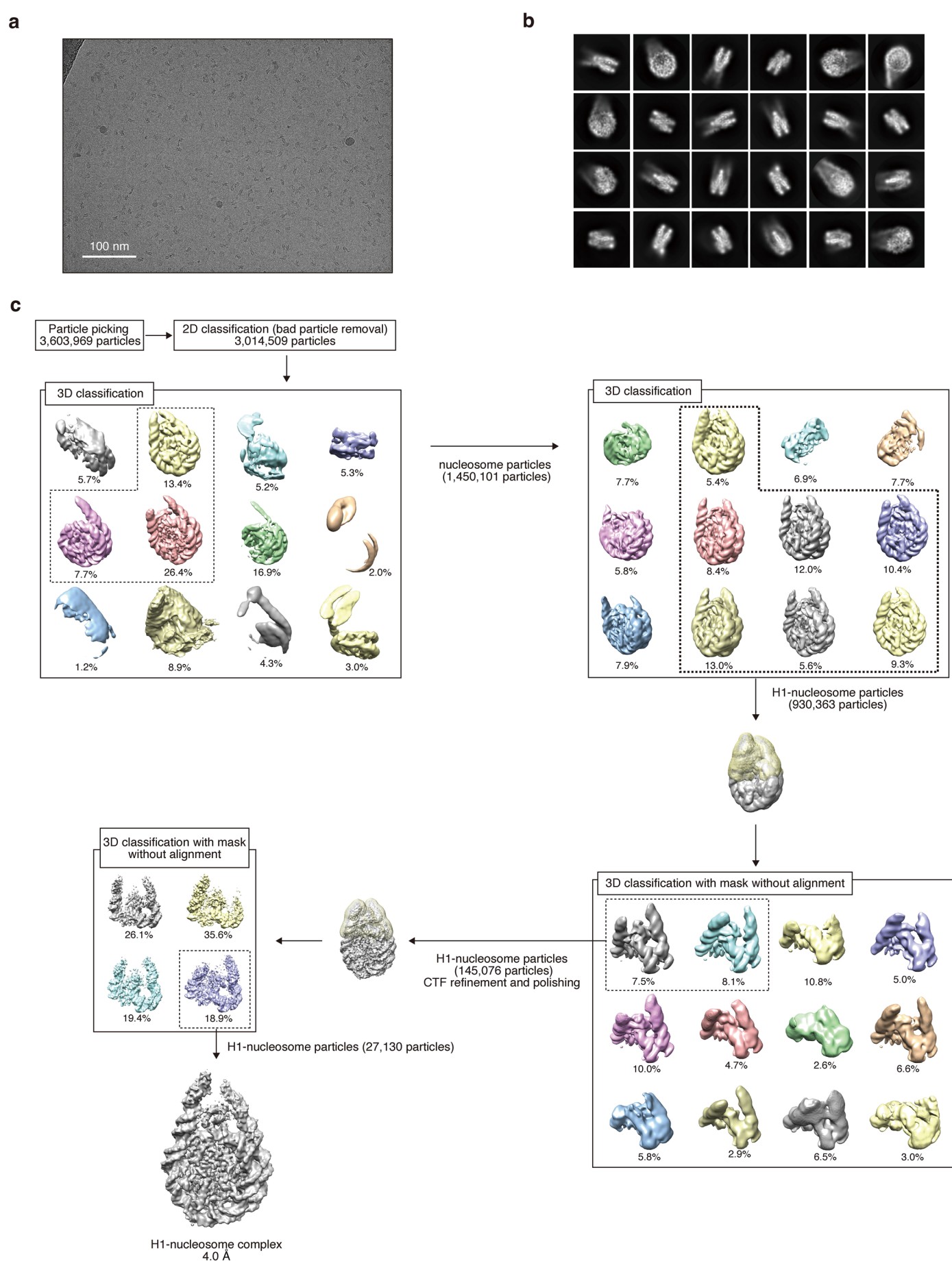

Extended Data Fig. 8 | See next page for caption.

**Extended Data Fig. 8 | Cryo-EM data collection and image processing of the H1-nucleosome complex.** (**a**) Representative micrograph of the H1-nucleosome complex. 8340 micrographs were collected. (**b**) Representative 2D classes of the H1-nucleosome complex. (**c**) Workflow of the image processing.

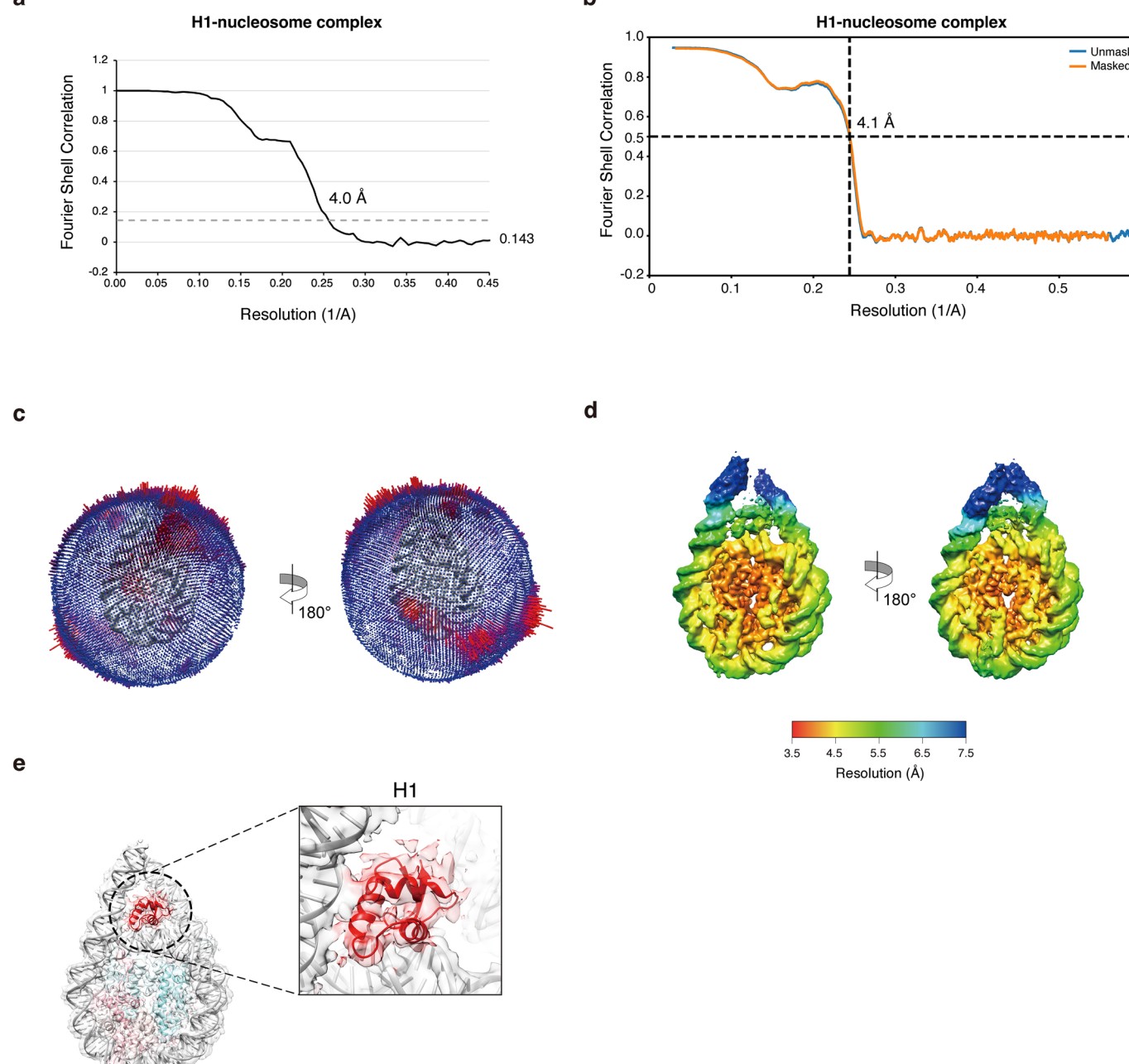

**Extended Data Fig. 9 | Quality of the H1-nucleosome complexes.** (**a**) Fourier Shell Correlation (FSC) curve for the H1-nucleosome complex. The resolution of the structure was estimated to be 4.0 Å by an FSC = 0.143. (**b**) Map to model FSC curves for the H1-nucleosome complex. (**c**) Euler angular distribution of the structure of the H1-nucleosome complex. (**d**) Local resolution map of the structure of the H1-nucleosome complex. (**e**) Close-up view of the H1 globular domain model in the nucleosome fitted with the EM density map.

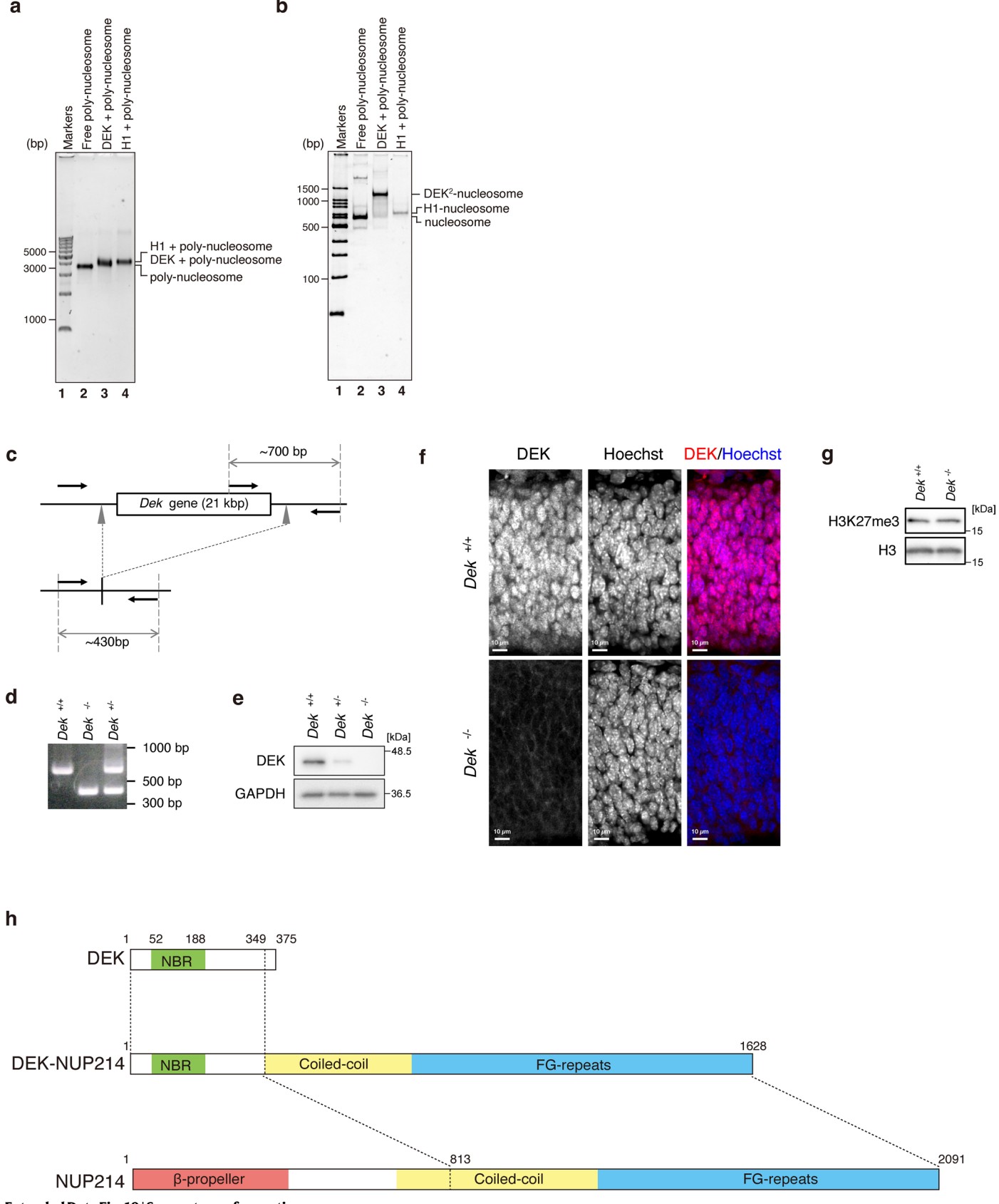

**Extended Data Fig. 10 | See next page for caption.**

**Extended Data Fig. 10 | The poly-nucleosomes used for the AFM analysis, H3K27me3 in *Dek* knockout mice, and the chimeric DEK-NUP214 protein.**
(**a**) The reconstituted poly-nucleosomes with 12 repeats of the 208 bp Widom 601 sequence DNA, with or without DEK and H1, were analyzed by agarose-gel electrophoresis with ethidium bromide staining. This experiment was conducted once. (**b**) The poly-nucleosomes with or without DEK or H1 were digested to mono-nucleosomes by *Sca*I. The resulting mono-nucleosomes were analyzed by native-PAGE with ethidium bromide staining. This experiment was conducted once. (**c**) Schematic representation of the *Dek* gene deletion strategy. (**d**) Genotyping PCR analysis for confirmation of *Dek* deletion. This experiment was conducted with more than 10 mice in each condition at postnatal day (P) 10, and repeated more than ten times with consistent results.

(**e**) Western blot analysis of the DEK protein in wild-type and *Dek* knockout mice. This experiment was conducted once with 1 mouse in each condition at E11.5.
(**f**) Immunohistochemistry analysis of the DEK protein in wild-type and *Dek* knockout mouse embryos at E11.5. This experiment was performed with 3 mice in each condition at E11.5, and reproduced in four independent experiments.
(**g**) Western blot analysis of H3K37me3 in the neocortexes of wild-type and *Dek* knockout mouse E11.5 embryos. This experiment was performed with 4 mice in each condition at E11.5 and reproduced in four independent experiments. (**h**) Schematic representation of the human DEK, DEK-NUP214, and human NUP214 proteins. DEK residues 1-349 are maintained in the chimeric DEK-NUP214. NBR: Nucleosome-binding region.

# Reporting Summary

## Statistics

For all statistical analyses, confirm that the following items are present in the figure legend, table legend, main text, or Methods section.

| n/a | Confirmed | |
|---|---|---|
| ☐ | ☒ | The exact sample size (*n*) for each experimental group/condition, given as a discrete number and unit of measurement |
| ☐ | ☒ | A statement on whether measurements were taken from distinct samples or whether the same sample was measured repeatedly |
| ☐ | ☒ | The statistical test(s) used AND whether they are one- or two-sided<br>*Only common tests should be described solely by name; describe more complex techniques in the Methods section.* |
| ☐ | ☒ | A description of all covariates tested |
| ☐ | ☒ | A description of any assumptions or corrections, such as tests of normality and adjustment for multiple comparisons |
| ☐ | ☒ | A full description of the statistical parameters including central tendency (e.g. means) or other basic estimates (e.g. regression coefficient) AND variation (e.g. standard deviation) or associated estimates of uncertainty (e.g. confidence intervals) |
| ☐ | ☒ | For null hypothesis testing, the test statistic (e.g. *F*, *t*, *r*) with confidence intervals, effect sizes, degrees of freedom and *P* value noted<br>*Give P values as exact values whenever suitable.* |
| ☒ | ☐ | For Bayesian analysis, information on the choice of priors and Markov chain Monte Carlo settings |
| ☒ | ☐ | For hierarchical and complex designs, identification of the appropriate level for tests and full reporting of outcomes |
| ☒ | ☐ | Estimates of effect sizes (e.g. Cohen's *d*, Pearson's *r*), indicating how they were calculated |

*Our web collection on statistics for biologists contains articles on many of the points above.*

## Software and code

Policy information about availability of computer code

| Data collection | AcquireMP ver. 2023 R1.1, SerialEM ver.3, EPU 3.1, NanoWizard IIR Instrument (JPK) |
|---|---|
| Data analysis | Relion 3.1.4, Relion 4.0.0, MotionCor2 1.3.2 and 1.4.0, gctf v1.18 and v1.60, ctffind-4.1.14, UCSF ChimeraX 1.2 and 1.6, ISOLDE 1.2 and 1.6, Coot 0.9.3, PHENIX 1.21rc1,UCSF Chimera 1.14, PyMOL, 2.5.5, ImageJ 1.52q, DiscoverMP ver. 2023 R1.2, gwyddion 2.64, RStudio 2022.07.2 +576, FASTX-Toolkit 0.0.14, cutadapt 2.6, bowtie2 2.3.5.1, bedtools2 2.29.0, samtools 1.9, f-seq 1.84, hisat2 2.1.0, featureCounts 1.6.0, ngsplot 2.63, MolProbity in PHENIX 1.21rc1 |

For manuscripts utilizing custom algorithms or software that are central to the research but not yet described in published literature, software must be made available to editors and reviewers. We strongly encourage code deposition in a community repository (e.g. GitHub). See the Nature Portfolio guidelines for submitting code & software for further information.

## Data

Policy information about availability of data

All manuscripts must include a data availability statement. This statement should provide the following information, where applicable:
- Accession codes, unique identifiers, or web links for publicly available datasets
- A description of any restrictions on data availability
- For clinical datasets or third party data, please ensure that the statement adheres to our policy

The cryo-EM reconstructions and atomic models of the DEK2, DEK1 - and H1-nucleosomes have been deposited in the Electron Microscopy Data Bank and the

## Research involving human participants, their data, or biological material

Policy information about studies with human participants or human data. See also policy information about sex, gender (identity/presentation), and sexual orientation and race, ethnicity and racism.

| | |
|---|---|
| Reporting on sex and gender | N/A |
| Reporting on race, ethnicity, or other socially relevant groupings | N/A |
| Population characteristics | N/A |
| Recruitment | N/A |
| Ethics oversight | N/A |

Note that full information on the approval of the study protocol must also be provided in the manuscript.

# Field-specific reporting

Please select the one below that is the best fit for your research. If you are not sure, read the appropriate sections before making your selection.

☒ Life sciences  ☐ Behavioural & social sciences  ☐ Ecological, evolutionary & environmental sciences

For a reference copy of the document with all sections, see nature.com/documents/nr-reporting-summary-flat.pdf

# Life sciences study design

All studies must disclose on these points even when the disclosure is negative.

| | |
|---|---|
| Sample size | Sample size calculation of AFM analyses was not conducted, but Particles were collected in numbers comparable to well established experimental approaches.(Wang et al., Biophys. J., 2009). Sample sizes of genome study were chosen as standard in the field (Hirabayashi et al., Nauron, 2009; Eto et al., Nature Communications, 2020). The replicate numbers were based on estimation from previous studies. |
| Data exclusions | During cryo-EM analyses, bad particles were excluded as shown Extended Data Figures. Other than the cryo-EM analyses, no data was excluded from the analysis. |
| Replication | The reproducibility of the findings were confirmed by performing at least two independent experiments. All replicated experiments were performed successfully. Cryo-EM analyses were conducted once because the final map already represented the average of a large number of images. |
| Randomization | Data was not randomized during biochemical analyses. Cryo-EM processing software Relion randomly splits particles into two different half-maps during 3D reconstruction in order to calculate resolution of the 3D volume. Randomization was not relevant to biochemical experiments. For genome study, all sample allocation were randomized. |
| Blinding | Since there was no subjective allocation in our investigation, blinding was irrelevant to this study |

# Reporting for specific materials, systems and methods

We require information from authors about some types of materials, experimental systems and methods used in many studies. Here, indicate whether each material, system or method listed is relevant to your study. If you are not sure if a list item applies to your research, read the appropriate section before selecting a response.

## Materials & experimental systems

| n/a | Involved in the study |
|---|---|
| ☐ | ☒ Antibodies |
| ☐ | ☒ Eukaryotic cell lines |
| ☒ | ☐ Palaeontology and archaeology |
| ☐ | ☒ Animals and other organisms |
| ☒ | ☐ Clinical data |
| ☒ | ☐ Dual use research of concern |
| ☒ | ☐ Plants |

## Methods

| n/a | Involved in the study |
|---|---|
| ☐ | ☒ ChIP-seq |
| ☒ | ☐ Flow cytometry |
| ☒ | ☐ MRI-based neuroimaging |

## Antibodies

| | |
|---|---|
| Antibodies used | anti-H3K27me3 antibody (Cell Signaling #9733) used at a 1:1000 dilution.<br>anti-H3K27me3 antibody (1E7) used at a 1:1000 dilution.<br>anti-Rabbit IgG, HRP-Linked F(ab')2 Fragment Donkey (Cytiva #NA9340) used at a 1:3000 dilution.<br>anti-Mouse IgG HRP Linked F(ab')2 Fragment (Cytiva #NA9310) used at a 1:5000 dilution.<br>anti-H2B antibody (Cell Signaling #2934) used at a 1:1000 dilution.<br>anti-rabbit IgG conjugated with Alexa Fluor 488 (Jackson Immuno Research Laboratories, Inc. #111-545-144) used at a 1:500 dilution.<br>anti-mouse IgG conjugated with Alexa Fluor 647 (Invitrogen, #A32728) used at a 1:5000 dilution.<br>H2AK119ub1 (Cell Signaling #8240S) used at a 1:100 dilution.<br>anti-DEK (BD, 610948) used at a 1:500 dilution.<br>anti-GAPDH (Cell Signaling, 2118S) used at a 1:500 dilution.<br>anti-H3 (Active Motif, 39763) used at a 1:5000 dilution.<br>anti–mouse Alexa Fluor 555 (Thermo Fisher Scientific, A-31570) used at a 1:1000 dilution. |
| Validation | anti-H3K27me3 antibody (Cell Signaling #9733). Western blot of various cell lines was performed. (https://www.cellsignal.com/products/primary-antibodies/tri-methyl-histone-h3-lys27-c36b11-rabbit-mab/9733)<br>anti-H3K27me3 antibody (1E7). ELISA using the valious H3 N-terminus tail peptides was performed. (https://academic.oup.com/nar/article/39/15/6475/1023574?login=false)<br>anti-DEK antibody (BD #610948). Western blot of a Jurkat cell lysate was performed. (https://www.bdbiosciences.com/en-br/products/reagents/microscopy-imaging-reagents/immunofluorescence-reagents/purified-mouse-anti-human-dek.610948)<br>H2AK119ub1 (Cell Signaling #8240S). Western blot of various cell lines was performed. (https://www.cellsignal.com/products/primary-antibodies/ubiquityl-histone-h2a-lys119-d27c4-xp-rabbit-mab/8240)<br>anti-GAPDH (Cell Signaling, 2118S). Western blot of various cell lines was performed. (https://www.cellsignal.jp/products/primary-antibodies/gapdh-14c10-rabbit-mab/2118)<br>anti-H3 (Active Motif, 39763). Western blot of a Hela cell lysate was performed. (https://www.activemotif.com/catalog/details/39763) |

## Eukaryotic cell lines

Policy information about cell lines and Sex and Gender in Research

| | |
|---|---|
| Cell line source(s) | HEK293: provided by Dr. Tetsu Akiyama, the University of Tokyo |
| Authentication | The authors declare that the cell lines were authenticated based on their morphology |
| Mycoplasma contamination | The cell lines were not tested for mycoplasma contamination. |
| Commonly misidentified lines (See ICLAC register) | HEK293 |

## Animals and other research organisms

Policy information about studies involving animals; ARRIVE guidelines recommended for reporting animal research, and Sex and Gender in Research

| | |
|---|---|
| Laboratory animals | JCL:ICR (CLEA Japan) and Slc:ICR (Kapan SLC), embryonic day 11 to 12 |
| Wild animals | No wild animals |
| Reporting on sex | No data analyses based on sex differences. Pooled sample of male and female were used. |
| Field-collected samples | No field-collected samples |
| Ethics oversight | Animal Care and Use Committee of The University of Tokyo (approval numbers: P25-8 and P30-4 in the Graduate School of Pharmaceutical Sciences, and 0421 and A2022IQB001 in the Institute for Quantitative Biosciences) |

Note that full information on the approval of the study protocol must also be provided in the manuscript.

## Plants

| | |
|---|---|
| Seed stocks | *Report on the source of all seed stocks or other plant material used. If applicable, state the seed stock centre and catalogue number. If plant specimens were collected from the field, describe the collection location, date and sampling procedures.* |
| Novel plant genotypes | *Describe the methods by which all novel plant genotypes were produced. This includes those generated by transgenic approaches, gene editing, chemical/radiation-based mutagenesis and hybridization. For transgenic lines, describe the transformation method, the number of independent lines analyzed and the generation upon which experiments were performed. For gene-edited lines, describe the editor used, the endogenous sequence targeted for editing, the targeting guide RNA sequence (if applicable) and how the editor was applied.* |
| Authentication | *Describe any authentication procedures for each seed stock used or novel genotype generated. Describe any experiments used to assess the effect of a mutation and, where applicable, how potential secondary effects (e.g. second site T-DNA insertions, mosiacism, off-target gene editing) were examined.* |

## ChIP-seq

### Data deposition

☒ Confirm that both raw and final processed data have been deposited in a public database such as GEO.

☒ Confirm that you have deposited or provided access to graph files (e.g. BED files) for the called peaks.

| | |
|---|---|
| Data access links *May remain private before publication.* | Database: the DNA Data Bank of Japan (DDBJ)<br>Raw sequence files: DRA016775 (DEK ChIP-seq), DRA016561 (H3K27me3 ChIP-seq), DRA016776 (H3K27me3 CUT&Tag), DRA016778 (H2AK119ub1 CUT&Tag), DRA016781 (ATAC-seq), and DRA016780 (RNA-seq).<br>Processed files : E-GEAD-680 (DEK ChIP-seq, called peaks, bed files), E-GEAD-681 (DEK ChIP-seq, bigwig files), E-GEAD-681 (H3K27me3 CUT&Tag, called peaks, bed files), E-GEAD-682 (H2Aub CUT&Tag, called peaks, bed files), E-GEAD-683 (ATAC-seq, called peaks, bed files), E-GEAD-678 (H3K27me3 ChIP-seq, called peaks, bed files), E-GEAD-679 (H3K27me3 ChIP-seq, bigwig files), DRA019123 (DEK ChIP-seq using HEK293 cells), E-GEAD-874 ((DEK ChIP-seq using HEK293 cells, bigwig files) |
| Files in database submission | DEK ChIP-seq, rep 1<br>DEK ChIP-seq, rep 3<br>H3K27me3 CUT&Tag_Con, rep 1<br>H3K27me3 CUT&Tag_Con, rep 2<br>H2AK119ub1 CUT&Tag_Con, rep 1<br>H2AK119ub1 CUT&Tag_Con, rep 2<br>ATAC-seq_Con, rep 1<br>ATAC-seq_Con, rep 2<br>ATAC-seq_Con, rep 3<br>ATAC-seq_Con, rep 4<br>H3K27me3 CUT&Tag_Con, rep 1<br>H3K27me3 CUT&Tag_Con, rep 2<br>H3K27me3 CUT&Tag_Con, rep 3<br>DEK ChIP-seq using HEK293, rep 1<br>DEK ChIP-seq using HEK293, rep 2 |
| Genome browser session<br>(e.g. UCSC) | No longer available. |

### Methodology

| | |
|---|---|
| Replicates | Two (DEK ChIP-seq, H3K27me3 CUT&Tag, H2Aub CUT&Tag), three (H3K27me3 ChIP-seq), four (ATAC-seq) |
| Sequencing depth | 151 bp paired end (DEK ChIP-seq, H3K27me3 CUT&Tag, H2Aub CUT&Tag), 100 bp paired end (ATAC-seq), 50 bp single end(H3K27me3 ChIP-seq) |
| Antibodies | DEK (BD, 610948)<br>H3K27me3 (CST, 9733S)<br>H2Aub (CST, 8240S) |
| Peak calling parameters | Software: f-seq<br>Parameters: t5 (DEK ChIP-seq), t7 (ATAC-seq), t9 (H3K27me3, H2Aub CUT&Tag) |
| Data quality | Significant signals in positive control regions (Hoxa cluster for H3K27me3 and H2Aub, Actb for ATAC-seq) and lower signals in negative control regions (Actb for H3K27me3 and H2Aub and Hoxa cluster for ATAC-seq) |
| Software | FASTX-Toolkit 0.0.14, cutadapt 2.6, bowtie2 2.3.5.1, bedtools2 2.29.0, samtools 1.9, f-seq 1.84, hisat2 2.1.0, featureCounts 1.6.0, ngsplot 2.63 |

