## [Peer Review File · Nature Structural & Molecular Biology]

Structural insights into how DEK nucleosome binding facilitates H3K27 trimethylation in chromatin

Corresponding Author: Professor Hitoshi Kurumizaka

Version 0:

Decision Letter:

24th May 2024

Dear Professor Kurumizaka,

Thank you again for submitting your manuscript "Structural basis of nucleosome binding by DEK that facilitates H3K27 trimethylation in chromatin". I apologize for the delay in responding, which resulted from the difficulty in obtaining suitable referee reports, combined with editorial absences in the last weeks. Nevertheless, we now have comments (below) from the 3 reviewers who evaluated your paper. In light of those reports, we remain interested in your study and would like to see your response to the comments of the referees, in the form of a revised manuscript.

You will see that while the reviewers highly appreciate the insights provided into the role of DEK, Reviewer #1 requests expanding on how DEK stimulates the methyltransferase activity of PRC2, echoed by Reviewer #2. Additionally Reviewer #1 asks to expand on whether the monomeric or dimeric form is the functional state, and the role of the DEK C-terminus. Reviewer #2 further requests discussion of the effect of DEK on poly-nucleosome compaction, and Reviewer #3 requests DEK ChIP-Seq in human cells and expansion on the effect of DEK inhibition on H3K27me3 levels. Please be sure to address/respond to all concerns of the referees in full in a point-by-point response and highlight all changes in the revised manuscript text file.

We appreciate the requested revisions are extensive. We thus expect to see your revised manuscript within 3-4 months. If you cannot send it within this time, please let us know. We will be happy to consider your revision as long as nothing similar has been accepted for publication at NSMB or published elsewhere. Should your manuscript be substantially delayed without notifying us in advance and your article is eventually published, the received date would be that of the revised, not the original, version.

Reporting Summary:

Please note that all key data shown in the main figures as cropped gels or blots should be presented in uncropped form, with molecular weight markers. These data can be aggregated into a single supplementary figure. While these data can be displayed in a relatively informal style, they must refer back to the relevant figures. These data should be submitted with the last revision, prior to acceptance, but you may want to start putting it together at this point.

SOURCE DATA: we request that authors provide, in tabular form, the data underlying the graphical representations used in figures. This is to further increase transparency in data reporting, as detailed in this editorial (<http://www.nature.com/nsmb/journal/v22/n10/full/nsmb.3110.html>). Spreadsheets can be submitted in excel format. Only one (1) file per figure is permitted; thus, for multi-paneled figures, the source data for each panel should be clearly labeled in the Excel file; alternately the data can be provided as multiple, clearly labeled sheets in an Excel file. When submitting files, the title field should indicate which figure the source data pertains to. We encourage our authors to provide source data at the revision stage, so that they are part of the peer-review process.

We require deposition of coordinates (and, in the case of crystal structures, structure factors) into the Protein Data Bank with the designation of immediate release upon publication (HPUB). Electron microscopy-derived density maps and coordinate data must be deposited in EMDB and released upon publication. Deposition and immediate release of NMR chemical shift assignments are highly encouraged. Deposition of deep sequencing and microarray data is mandatory, and the datasets must be released prior to or upon publication. To avoid delays in publication, dataset accession numbers must be supplied with the final accepted manuscript and appropriate release dates must be indicated at the galley proof stage. Please find the complete NRG policies on data availability at <http://www.nature.com/authors/policies/availability.html>.

Link Redacted

Sincerely,
Sara

Sara Osman, Ph.D.
Senior Editor
Nature Structural & Molecular Biology

Referee expertise:

Referee #1: Structural biology, epigenetics

Referee #2: Chromatin, Cryo-EM

Referee #3: Epigenomics, cancer, gene regulation

Reviewers' Comments:

Reviewer #1:

Remarks to the Author:

DEK is a ubiquitous non-histone chromosomal protein known for its affinity towards supercoiled and cruciform DNA structures. Its involvement in critical cellular processes such as heterochromatin maintenance, transcriptional regulation, and DNA repair underscores its significance. However, the precise mechanisms underlying DEK's influence on chromatin

architecture and genome regulation have remained elusive. In this study, the authors make significant strides in understanding DEK's role. They demonstrate that DEK augments the methyltransferase activity of the Polycomb repressive complex 2 (PRC2), leading to an elevation in the facultative heterochromatin mark H3K27me3. Notably, DEK exhibits colocalization with mono-ubiquitinated H2AK119 (H2AK119ub) alongside H3K27me3 peaks. Given PRC2's partnership with PRC1 in stem cells to deposit H3K27me3 and H2AK119ub at facultative heterochromatin, it is inferred that DEK modulates gene expression by influencing these repressive histone marks around transcription start sites (TSSs). Furthermore, the authors elucidate the cryo-electron microscopy (cryo-EM) structure of the DEK-nucleosome complex. These structural analyses reveal DEK's binding to the nucleosome in both dimeric and monomeric forms through tripartite DNA binding coupled with acidic patch binding. Atomic force microscopy observations demonstrate that DEK induces chromatin compaction within the DEK-poly-nucleosome complex.

Overall, this study significantly advances our understanding of DEK's role in chromatin regulation and provides a foundation for further investigations into its molecular mechanisms and biological significance. I support its publication before the following concerns are addressed:

1. How does DEK stimulate the methyltransferase activity of Polycomb repressive complex 2? Does the DEK recruit the PRC2 to the nucleosome, or DEK mediate chromatin compaction and then facilitate PRC2 to methylate H3K27?
2. DEK can bind to nucleosomes in a dimeric or monomeric form, and the dimer is a major nucleosome-binding form. Which form is the biologically functional form of DEK?
3. The cryo-EM structure of the DEK-nucleosome complex revealed that the C-terminus of DEK does not participate in interactions with the nucleosome. Consistently, the regions of DEK essential for nucleosome binding are preserved in the DEK-NUP214 fusion protein. It would be intriguing to examine quantitative data or even delve into complex structures to compare the binding differences among DEK, DEK lacking the C-terminus, and the DEK-NUP124 fusion proteins.

Reviewer #2:

Remarks to the Author:

DEK is an abundant chromosomal protein whose function is little understood. This manuscript by Kujirai et al. presents the cryo-EM structure of DEK in complex with the nucleosome. The authors also identify that DEK can stimulate the activity of PRC2 in vitro and colocalizes with H3K27me3 mark in embryonic neural progenitor. These results provide very important insight into the structural basis of chromatin attachment of DEK and how it regulates gene activity. The data presented is solid, and the analyses are thorough. The manuscript is very well written and is appropriate for publication in NSMB. I have only a few minor concerns to be addressed:

1. The authors show that DEK significantly promotes the activity of PRC2 on poly-nucleosome. Does DEK also promote the activity of PRC2 on the 193bp nucleosome used in the structure determination? Could the authors explain or speculate how DEK binding to nucleosome can stimulate the activity of PRC2?
2. In Fig. 7, the authors show that DEK can further compact the poly-nucleosome. The free poly-nucleosome shown in panel B seems already compacted. Is magnesium ion used to prepare the poly-nucleosome? If so, could DEK compact poly-nucleosome array in the absence of magnesium ion like the linker histone H1 does?
3. In the second paragraph of Model building in Methods section, some of the references are mislabeled. Please check and update them.
4. The map-to-model FSC curves are not provided in the extended figures. It would be good if the authors could add them.

Reviewer #3:

Remarks to the Author:

In this manuscript, Kujirai et al. show that the oncoprotein DEK stimulates PRC2 activity and leads to H3K27me3 deposition. The work is very nicely performed, and this finding is an important conclusion worth publishing in NSMB. However, I also have some questions about the work.

1. The authors showed that DEK ChIP-Seq in mouse brain colocalizes with facultative heterochromatin. But what about in human? The authors very oddly jump from talking about mice to suddenly talking about human with their recombinant assay. It would connect the two sections together if the authors could also do DEK ChIP-Seq in human cells.
2. Does inhibition of PRC2 in mouse and human lead to abrogation of DEK H3K27me3 deposition?
3. I would imagine that inhibition or CRISPR removal of DEK would lead to loss of H3K27me3. However, in PMID: 33755722, the authors examined H3K27me3 levels in wild-type and DEK-depleted cells, but they did not show any clear difference in H3K27me3 levels (Figure 5A). What could be the reason for this disparity?
4. Is there any mass spectrometry data or other protein-protein interaction data showing that DEK binds to PRC2 in vivo?

Version 1:

Decision Letter:

Our ref: NSMB-A48760A

18th Nov 2024

Dear Professor Kurumizaka,

Thank you for submitting your revised manuscript "Structural basis of nucleosome binding by DEK that facilitates H3K27 trimethylation in chromatin" (NSMB-A48760A). It has now been seen by the original referees and their comments are below. The reviewers find that the paper has improved in revision, and therefore we are happy to accept it in principle in Nature Structural & Molecular Biology, pending minor revisions to satisfy the referees' final requests and to comply with our editorial and formatting guidelines. Please note that original Reviewer #1 was unavailable at this round and was substituted by Reviewer #4 who holds very similar scientific expertise.

We are now performing detailed checks on your paper and will send you a checklist detailing our editorial and formatting requirements in about 3 weeks. Please do not upload the final materials and make any revisions until you receive this additional information from us.

To facilitate our work at this stage, it is important that we have a copy of the main text as a word file. If you could please send along a word version of this file as soon as possible, we would greatly appreciate it; please make sure to copy the NSMB account (cc'ed above).

Sincerely,

Dimitris Typas
Senior Editor
Nature Structural & Molecular Biology
ORCID: 0000-0002-8737-1319

Reviewer #2 (Remarks to the Author):

The authors have addressed all my concerns in detail. I am satisfied with the revision. Given this, this work would be appropriate for publication in NSMB.

Reviewer #3 (Remarks to the Author):

My concerns have been addressed. I think this manuscript should be accepted.

Reviewer #4 (Remarks to the Author):

In this study, the authors make significant progress in elucidating DEK's role in chromatin regulation, demonstrating that DEK enhances the methyltransferase activity of the Polycomb repressive complex 2 (PRC2), resulting in an increase in the facultative heterochromatin mark H3K27me3. The authors have provided a satisfactory response to the comments by conducting the relevant experiments. I have no additional questions or concerns for the authors.

Version 2:

Decision Letter:

22nd Jan 2025

Dear Professor Kurumizaka,

We are now happy to accept your revised paper "Structural insights into how DEK nucleosome binding facilitates H3K27 trimethylation in chromatin" for publication as an Article in Nature Structural & Molecular Biology.

Your paper will be published online soon after we receive proof corrections and will appear in print in the next available issue. You can find out your date of online publication by contacting the production team shortly after sending your proof corrections.

Authors may need to take specific actions to achieve <https://www.springernature.com/gp/open-research/funding/policy-compliance-faqs> compliance with funder and institutional open access mandates. If your research is supported by a funder that requires immediate open access (e.g. according to <https://www.springernature.com/gp/open-research/plan-s-compliance> Plan S principles) then you should select the gold OA route, and we will direct you to the compliant route where possible. For authors selecting the subscription publication route, the journal's standard licensing terms will need to be accepted, including <https://www.springernature.com/gp/open-research/policies/journal-policies> self-archiving policies. Those licensing terms will supersede any other terms that the author or any third party may assert apply to any version of the manuscript.

Sincerely,

Dimitris Typas
Senior Editor
Nature Structural & Molecular Biology
ORCID: 0000-0002-8737-1319

Response to the editor

You will see that while the reviewers highly appreciate the insights provided into the role of DEK, Reviewer #1 requests expanding on how DEK stimulates the methyltransferase activity of PRC2, echoed by Reviewer #2. Additionally Reviewer #1 asks to expand on whether the monomeric or dimeric form is the functional state, and the role of the DEK C-terminus. Reviewer #2 further requests discussion of the effect of DEK on poly-nucleosome compaction, and Reviewer #3 requests DEK ChIP-Seq in human cells and expansion on the effect of DEK inhibition on H3K27me3 levels. Please be sure to address/respond to all concerns of the referees in full in a point-by-point response and highlight all changes in the revised manuscript text file.

Reply)

Thank you very much for providing their constructive comments. In the revised manuscript, we responded to these concerns in full, as described below.

Responses to the Reviewers

Reviewer #1

DEK is a ubiquitous non-histone chromosomal protein known for its affinity towards supercoiled and cruciform DNA structures. Its involvement in critical cellular processes such as heterochromatin maintenance, transcriptional regulation, and DNA repair underscores its significance. However, the precise mechanisms underlying DEK's influence on chromatin architecture and genome regulation have remained elusive. In this study, the authors make significant strides in understanding DEK's role. They demonstrate that DEK augments the methyltransferase activity of the Polycomb repressive complex 2 (PRC2), leading to an elevation in the facultative heterochromatin mark H3K27me3. Notably, DEK exhibits colocalization with mono-ubiquitinated H2AK119 (H2AK119ub) alongside H3K27me3 peaks. Given PRC2's partnership with PRC1 in stem cells to deposit H3K27me3 and H2AK119ub at facultative heterochromatin, it is inferred that DEK modulates gene expression by influencing these repressive histone marks around transcription start sites (TSSs). Furthermore, the authors elucidate the cryo-electron microscopy (cryo-EM) structure of the DEK-nucleosome complex. These structural analyses reveal DEK's binding to the nucleosome in both dimeric and monomeric forms through tripartite DNA binding coupled with acidic patch binding. Atomic force microscopy observations demonstrate that DEK induces chromatin compaction within the DEK-poly-nucleosome complex.

Overall, this study significantly advances our understanding of DEK's role in chromatin regulation and provides a foundation for further investigations into its molecular mechanisms and biological significance. I support its publication before the following concerns are addressed:

1. How does DEK stimulate the methyltransferase activity of Polycomb repressive complex 2? Does the DEK recruit the PRC2 to the nucleosome, or DEK mediate chromatin compaction and then facilitate PRC2 to methylate H3K27?

Reply)

Thank you for this insightful comment. In the revised manuscript, we performed the PRC2 binding assay with DEK, which revealed that DEK does not directly bind to PRC2. This suggests that DEK promotes PRC2 activation through chromatin compaction. We presented these new data in Extended Data Fig. 4c and 4d, and discussed them in the text (p.4, l.17-19).

2. DEK can bind to nucleosomes in a dimeric or monomeric form, and the dimer is a major nucleosome-binding form. Which form is the biologically functional form of DEK ?

Reply)

Our mass photometric analysis suggested that DEK does not form a dimer in solution, but two DEKs bound to one nucleosome may be a major form. To reconcile this contradiction, we performed an analysis with the DEK (102-105A) mutant, in which the residues forming the DEK-DEK interface were replaced by Ala. We then found that this DEK (102-105A) mutation does not affect its nucleosome binding. This suggested that two DEK molecules preferentially bind to one nucleosome, but the DEK-DEK interaction may not be crucial. Therefore, in the revised manuscript, we eliminated the term “DEK dimer” and instead described it as the nucleosome with two DEK molecules. These data are presented in the new Extended Data Fig. 10a, 10b, and 10c, and are discussed in the text (p.6, l.15-23).

3. The cryo-EM structure of the DEK-nucleosome complex revealed that the C-terminus of DEK does not participate in interactions with the nucleosome. Consistently, the regions of DEK essential for nucleosome binding are preserved in the DEK-NUP214 fusion protein. It would be intriguing to examine quantitative data or even delve into complex structures to compare the binding differences among DEK, DEK lacking the C-terminus, and the DEK-NUP124 fusion proteins.

Reply)

Thank you for this suggestion. In the revised manuscript, we prepared the DEK mutant lacking the C-terminus, and performed the nucleosome binding assay. We then found that the DEK C-terminal region somewhat reduces the nucleosome binding without affecting the nucleosome binding profile with one or two DEK molecules. These new data are presented in the new Extended Data Fig. 10h, 10i, and 10j, and the results are described in the text (p.7, l.29-36). We are quite interested in the DEK-NUP214 fusion protein, but have not successfully produced it yet. Therefore, it will be a fascinating project in the future.

Reviewer #2

DEK is an abundant chromosomal protein whose function is little understood. This manuscript by Kujirai et al. presents the cryo-EM structure of DEK in complex with the nucleosome. The authors also identify that DEK can stimulate the activity of PRC2 in vitro and colocalizes with H3K27me3 mark in embryonic neural progenitor. These results provide very important insight into the structural basis of chromatin attachment of DEK and how it regulates gene activity. The data presented is solid, and the analyses are thorough. The manuscript is very well written and is appropriate for publication in NSMB. I have only a few minor concerns to be addressed:

1. The authors show that DEK significantly promotes the activity of PRC2 on poly-nucleosome. Does DEK also promotes the activity of PRC2 on the 193bp nucleosome used in the structure determination? Could the authors explain or speculate how DEK binding to nucleosome can stimulate the activity of PRC2?

Reply)

Thank you very much for this insightful comment. According to this reviewer’s suggestion, we tested the PRC2 activity in the presence of the mono-nucleosome containing the 193bp DNA or 208bp DNA, which is a repeating unit of the poly-nucleosome DNA, instead of the poly-nucleosome. We then found that the PRC2-mediated H3K27me3 deposition was not

efficient with the mono-nucleosomes as compared to the poly-nucleosome. This suggested that the higher order chromatin conformation modulated by DEK, but not the mono-nucleosome level conformation, may stimulate the H3K27me3 deposition. These new 208bp nucleosome data are presented in the new Extended Data Fig. 4b, and are described in the text (p.5, 115-17). The 193bp nucleosome data are attached here for the reviewer.

2. In Fig. 7, the authors show that DEK can further compact the poly-nucleosome. The free poly-nucleosome shown in panel B seems already compacted. Is magnesium ion used to prepare the poly-nucleosome? If so, could DEK compact poly-nucleosome array in the absence of magnesium ion like the linker histone H1 does?

Reply)

We performed the AFM analyses in the absence of magnesium ion. The results suggested that DEK could compact the poly-nucleosome in a similar manner to linker histone H1. As this reviewer pointed out, poly-nucleosome compaction is induced by linker histone H1 in the absence of magnesium ion. Therefore, we performed the AFM analysis with the poly-nucleosome in the presence of histone H1 under the present experimental conditions, and found that H1 drastically compacted the poly-nucleosome. These new data are presented in the new Fig. 7b and 7c, and the results are discussed in the text (p.8, 128-31).

3. In the second paragraph of Model building in Methods section, some of the references are mislabeled. Please check and update them.

Reply)

We corrected them accordingly.

4. The map-to-model FSC curves are not provided in the extended figures. It would be good if the authors could add them.

Reply)

We added them accordingly.

Reviewer #3:

Remarks to the Author:

In this manuscript, Kujirai et al. show that the oncoprotein DEK stimulates PRC2 activity and leads to H3K27me3 deposition. The work is very nicely performed, and this finding is an important conclusion worth publishing in NSMB. However, I also have some questions about the work.

1. The authors showed that DEK ChIP-Seq in mouse brain colocalizes with facultative heterochromatin. But what about in human? The authors very oddly jump from talking about

mice to suddenly talking about human with their recombinant assay. It would connect the two sections together if the authors could also do DEK ChIP-Seq in human cells.

Reply)

Thank you for this suggestion. Accordingly, we performed the DEK ChIP-Seq in human cells, and consistent results were obtained. These new data are presented in the new Extended Data Fig. 3, and are described in the text (p.4,l36-37).

2. Does inhibition of PRC2 in mouse and human lead to abrogation of DEK H3K27me3 deposition?

Reply)

PRC2 is the sole methyltransferase responsible for the deposition of H3K27me3 (PMID: 29483650). Therefore, the inhibition of PRC2 may lead to the loss of all H3K27me3, as reported previously. We have not yet conducted specific experiments to assess whether the inhibition of PRC2 in mouse and human cells leads to the abrogation of DEK-associated H3K27me3 deposition. To answer this question, we will need to design and implement a novel experimental setup tailored for PRC2 inhibition. These experiments are planned for future work, once the experimental setup is established.

3. I would image that inhibition or CRISPR removal of DEK would lead to loss of H3K27me3. However, in PMID: 33755722, the authors examined H3K27me3 levels in wild-type and DEK-depleted cells, but they did not show any clear difference in H3K27me3 levels (Figure 5A). What could be the reason for this disparity?

Reply)

Thank you for your insightful comment. In our experiments, we found that H3K27me3 levels were not substantially affected in *Dek* knockout mice, consistent with the report (PMID: 33755722). This discrepancy may be attributed to the functional redundancy between DEK and linker histone H1. In our revised experiments, we demonstrated that PRC2 activation by DEK can be mediated through poly-nucleosome compaction. Supporting this, we showed that H1 induces poly-nucleosome compaction under the experimental conditions used in this study. We have included these new data in the revised manuscript as the new Fig. 7b and 7c, and discussed these findings in the text (p.8,l.28-31).

4. Is there any mass spectrometry data or other protein-protein interaction data showing that DEK binds to PRC2 in vivo?

Reply)

To test whether DEK directly binds to PRC2, we performed a pull-down assay. We then found that DEK does not directly bind to PRC2. This suggested that DEK binding modulates the poly-nucleosome condensation and promotes the H3K27me3 deposition by PRC2. Therefore, this provides important information to reveal the mechanism by which DEK stimulates the PRC2-mediated H3K27me3 deposition in chromatin. These new data are presented in the new Extended Data Fig. 4c and 4d, and the results are described in the text (p.5,l.17-19).